# Low cost exoskeleton manipulator using bidirectional triboelectric sensors enhanced multiple degree of freedom sensory system

Minglu Zhu[1,2,3,4], Zhongda Sun [1,2,3,4], Tao Chen [5✉] & Chengkuo Lee [1,2,3,4,6✉]

Rapid developments of robotics and virtual reality technology are raising the requirements of more advanced human-machine interfaces for achieving efficient parallel control. Exoskeleton as an assistive wearable device, usually requires a huge cost and complex data processing to track the multi-dimensional human motions. Alternatively, we propose a triboelectric bi-directional sensor as a universal and cost-effective solution to a customized exoskeleton for monitoring all of the movable joints of the human upper limbs with low power consumption. The corresponding movements, including two DOF rotations of the shoulder, twisting of the wrist, and the bending motions, are detected and utilized for controlling the virtual character and the robotic arm in real-time. Owing to the structural consistency between the exoskeleton and the human body, further kinetic analysis offers additional physical parameters without introducing other types of sensors. This exoskeleton sensory system shows a great potential of being an economic and advanced human-machine interface for supporting the manipulation in both real and virtual worlds, including robotic automation, healthcare, and training applications.

[1] Department of Electrical & Computer Engineering, National University of Singapore, Singapore 117576, Singapore. [2] National University of Singapore Suzhou Research Institute (NUSRI), Suzhou Industrial Park, Suzhou 215123, China. [3] Singapore Institute of Manufacturing Technology and National University of Singapore (SIMTech-NUS) Joint Lab on Large-area Flexible Hybrid Electronics, National University of Singapore, Singapore 117576, Singapore. [4] Center for Sensors and MEMS, National University of Singapore, Singapore 117576, Singapore. [5] Jiangsu Provincial Key Laboratory of Advanced Robotics, School of Mechanical and Electric Engineering, Soochow University, Suzhou 215123, China. [6] NUS Graduate School - Integrative Sciences and Engineering Program (ISEP), National University of Singapore, Singapore 119077, Singapore. ✉email: chent@suda.edu.cn; elelc@nus.edu.sg

Nowadays, robotics is becoming the indispensable technology for the future intelligent living environment and human society. The specialized robots with great precision provide significant assistance or even replace human's duties in many fields[1–3], such as medical operation, industrial automation, and tasks in extreme environments, etc. Noticeably, exoskeleton, as an emerging technique for the fusion of humans and robotics, is showing its own advantages in rehabilitation and operation assistance[4]. Except for the rigid structure, the continuous research efforts realize the soft exoskeleton as a suit for improving the comfortability[5]. On the other hand, to enhance the intelligence for both robots and exoskeletons, the development of human–machine interfaces (HMIs) equipped with diversified sensors is the mainstream technology to realize the parallel control for the advanced industrial automation and the interaction in virtual spaces[6–8].

Under 5 G and artificial intelligence of things (AIoT) infrastructure, the advancements in robotic and virtual reality/augmented reality (VR/AR) technologies offer more conveniences and better experiences to our life[9–11]. As the key device which links human to the digital world, HMIs show the burgeoning demand of more advanced solutions for realizing the dexterous and accurate manipulations of robots in real-world and virtual objects in the cyber world. Smart and vast distributed sensors are revolutionizing the whole society by boosting the capability, and lowering the cost of the sensory systems[12,13]. Eventually, all humans can be benefited via the seamless integration among the digitized bodies in the future AIoT living environment[14]. Currently, video and voice recognition are widely applied as the advanced HMIs compared to conventional controllers[15,16], but there is the privacy concern issue. On the other hand, wearable HMIs play a significant role by monitoring the signals intimately from the human body[17]. In terms of physical sensing techniques, several approaches are adopted in commercial products, including inertial sensors, resistive sensors, capacitive sensors[18,19], etc. Those inertial sensor-based HMIs usually offer high sensitivity for motion detections, and the resistive sensors possess the capabilities of both strain and force sensing. Eventually, glove-based HMIs with inertial and resistive sensors were reported frequently[20,21]. However, these sensing mechanisms also suffer several drawbacks, such as the complex signal processing of massive amount of data from inertial sensors and the temperature dependences of data from resistive sensors. Moreover, most of these sensing techniques consume sizable electrical power in the sensor operation and signal processing. This power consumption issue becomes more severe for the distributed sensory network with a large number of sensors.

In addition to the aforementioned commercialized HMIs, the newly developed flexible wearable sensor and e-skins offer promising technology to further enhance the intelligence and functionality of HMIs[22–28]. For instance, a dense array of capacitive tactile finger sensors was fabricated with a temperature sensing function[29]. An ultra-sensitive resistive strain gauge was reported for making a textile-based sensor-integrated sleeve for hand motion detection[30]. A skin patch with a multifunctional sensor array, including pressure, temperature, humidity, light, magnetic field, etc., was reported[31]. A hybridized sensor using piezoresistive and thermoelectric effects was realized to provide self-powered pressure and temperature sensing of high sensitivity[32]. Bai et al. developed a stretchable distributed fiber-optic sensory glove with the elastomeric lightguides which can produce continuum or discrete chromatic patterns for detecting the finger motions[33]. In the meantime, triboelectric and piezoelectric sensors featured with the self-generated signals are drawing increased attention, due to the lower power consumption of sensing elements compared to conventional devices[34–36], especially for the scenarios of

the massive distributed sensory network. Triboelectric sensors have wide options in terms of fabrication technology and materials[37–39], and the diverse mechanical stimuli including pressure, rotation, vibrations, etc., can be converted into electric outputs. It exerts the unique advantage in developing customizable multifunctional wearable sensors[40,41]. Several sensory gloves were presented with the arch-shaped or dome-shaped sensors via contact and separation working mode, and the grating patterned strip as the sliding mode sensor[42,43]. Additionally, the machine learning techniques also help to interpret the sensing information while maintaining the minimalistic design[44–47]. The unified sensing mechanism for various physical motions will reduce the signal processing complexity and the computing power, and provide ease of maintenance too.

In terms of the wearable platforms for supporting the fabricated sensors, the commonly adopted approaches include clothes, footwear, gloves, patches, etc. which possess great comfortability via the flexible designs[48–50]. However, the signal qualities, the sensitivities, and the accuracies become problematic for the measurement of several physical parameters, such as the rotation and the spatial positions of the joints. Exoskeletons, as the emerging wearable devices, are frequently developed as assistive equipment as mentioned before[51,52]. These exoskeleton suits mainly perform the actuation functions to enhance the mechanical output power of the human body for heavy tasks[53]. However, there are few researches focusing on new sensors for the exoskeleton enabled human–machine interactions[2,54,55]. Owing to the good consistency between the exoskeleton and the human body structure, the sensory information usually offers better projection of the spatial position and the motion dynamics, compare to those distributed flexible sensors. As to the rehabilitation of stroke patients, a customizable exoskeleton is necessary to monitor the static and dynamic information during the practice motions for evaluation and risk assessment[55]. The conventional inertial sensors enabled exoskeletons will generate a huge amount of data which consumes much higher computing power[56]. The vision-based HMIs can be adopted to analyze the body motion as well, but they usually require a clean space to avoid the block of the vision.

In this paper, we report a kind of triboelectric bidirectional (TBD) sensing mechanism integrated with an exoskeleton system for capturing and projecting the motions of the human arm and finger into the robotic arm or virtual space, as depicted in Fig. 1a (also see Supplementary Fig. 1). Further with the aid of kinetic analysis, the multiple physical parameters can be estimated through the data from the TBD sensors featured with the minimalistic design. Figure 1a shows the minimalistic design strategy of our approach to significantly reduce the complexity of the sensing system, as well as the computing power. The rotational TBD (RTBD) sensor consists of three major components, a shaft, a fly ring with the grating pattern of poly-tetra-fluoroethylene (PTFE) layers, and a bistable switch with two electrodes, which are in charge of detecting clockwise and counterclockwise rotation, respectively. The liner TBD sensor includes a switch fixture, a flexible substrate with the grating pattern of PTFE, and a bistable switch. According to the triboelectric sliding sensing mode, the TBD sensor can detect bidirectional rotation angles using the outputs from a single grating pattern of PTFE (Fig. 1b). Furthermore, a set of customized exoskeleton arms is 3D printed to enable the setup of TBD sensors for achieving the motion detection of multiple joints and multiple degrees of freedom (DOFs) with comparable resolutions which are suitable for the normal control. Together with the finger-based HMI, the motion projection of the entire arm/hand can be fulfilled using the facile-designed triboelectric sensors completely, owing to the utilization of the various triboelectric operation mode. In general, the

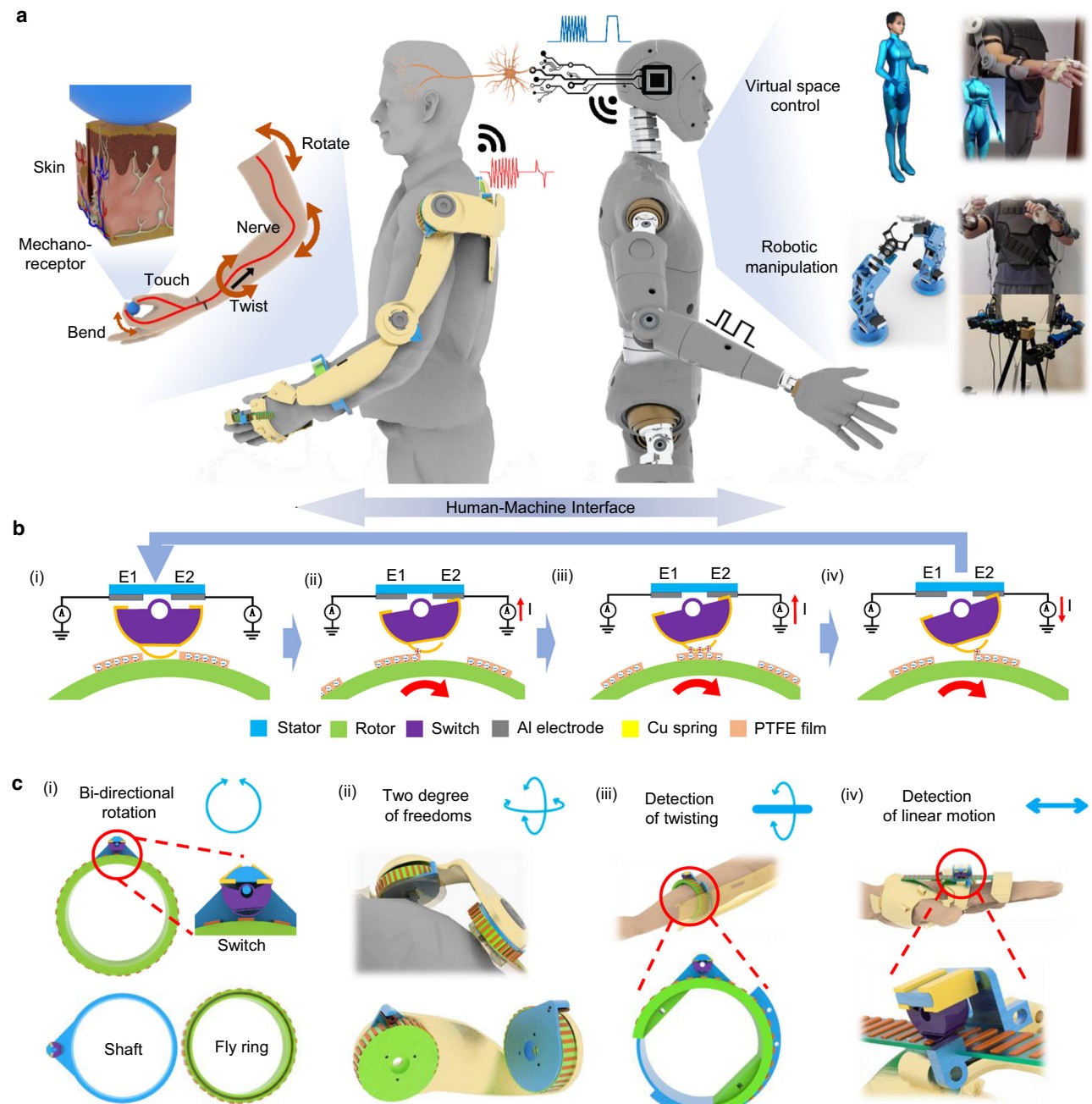

**Fig. 1 Triboelectric bidirectional (TBD) sensor-integrated exoskeleton system. a** Schematics of exoskeleton sensory system for realizing the manipulation in virtual space and robotics. **b** Working principle of TBD sensor. **c** Key functions for achieving the multidimensional motion sensing of the upper limb, (i) basic bidirectional rotation sensing, (ii) bidirectional rotation sensing with two degree of freedoms (DOFs), (iii) bidirectional detection of twisting motion of the wrist, and (iv) bidirectional linear motion sensing for finger bending. Photo credit: Minglu Zhu, National University of Singapore.

proposed exoskeleton sensory system offers a low-cost, energy-saving, and universal solution for realizing the tracking of body motions, and the estimation of multiple physical parameters, which provide a great assistance to the related researches in the parallel control applications of robotic automation, digital twin, training, and healthcare fields. The economic and facile designed devices are highly desired for accelerating the development in this field for bringing more conveniences to humans.

## Results

**Designs of rotational and linear TBD sensor.** A RTBD sensor has three main components made by 3D printing, including a

shaft, a fly ring, and a bistable switch. As shown in Fig. 1c(i), it is featured with a ring-shaped structure and a platform/holder as the switch fixture at the corner. A semicylinder stand is printed on the platform for the attachment of two electrodes of the switch. Another ring with the groove is designed to be the fly ring to place over the shaft for rotating, the outer surface is attached with the grating pattern using PTFE layers. Next, a semicylinder-shaped switch is applied and fixed onto the platform of the shaft (in between the ring and the stand) by a screw, while maintaining the rotation capability. In the completed assembly, the curved surface of switch should keep sliding across the grating pattern of the fly ring during rotation and generating the triboelectric

output for the electrodes to extract. In order to ensure the full contact between these two surfaces, while avoiding the jam phenomenon of two rigid moving parts in case they are pushed toward each other too close, the curved side of switch is redesigned with a flat surface, and a copper spring is used as a complementary part to attached onto the flat surface. Hence, the spring acts as a relative positive triboelectric material and a deformable contact surface against the relative negative PTFE grating pattern to ensure the smoothness of rotation without sacrificing the firmness of contacting. Moreover, the metallic foil helps to lead the output to the backside of switch, where a gap is intentionally designed in between the backside of switch and the stand with two electrodes. During the rotation of the fly ring, the sliding action can lead to the rotation of semicylinder switch and make one edge of the backside to contact with the corresponding electrode on the stand under a specific rotating direction, i.e., clockwise or counterclockwise. The triboelectric outputs are then extracted out for processing. By customizing this RTBD sensor for the corresponding parts of the upper limb, the multidimensional sensing can be realized, such as the detection of two DOFs motions (Fig. 1c(ii)), and the detection of the twisting of the wrist (Fig. 1c(iii)). The diameter of the ring-shaped RTBD sensors vary from 7 to 8.5 cm based on the dimensions of the platform on the exoskeleton, and the thickness is maintained as 2 cm.

As illustrated in Fig. 1c(iv), a linear TBD (LTBD) sensor, i.e., the finger sensor, consists of a holder as the switch fixture, a switch, and a flexible fluorinated ethylene propylene (FEP) strip with the similar PTFE based grating pattern. Unlike the RTBD sensor, the LTBD sensor does not have a ring-shaped structure. The switch holder can be installed onto the palm case with two screws, and the end of the FEP strip is fixed onto the finger case. Another end of the strip is, then, inserted into the gap between the switch and the switch holder.

**Working principle and sensing mechanism**. In Fig. 1b, based on the triboelectric theorem, as the copper spring sliding across the grating pattern made by PTFE layers on the fly ring, the variation of contacted electrification area will result the charge transfer through the connected external circuit, in order to neutralize the contact surface potential between PTFE layer with more electronegativity and copper spring with less electronegativity (see Supplementary Fig. 2). When the copper spring slide across each PTFE layer (Fig. 1b(ii–iv)), the charges will flow back and forth once. Hence, the whole grating pattern of PTFE layers can repetitively induce the output with pulse waveform for the switch during the rotation, so that the rotation status can be monitored. Besides, two electrodes on the stand for identifying the clockwise and counterclockwise rotations are named as E1 and E2. As the spring on the switch touches the rotor, the shear force caused by rotation will twist the switch to deflect/rotate toward either clockwise or counterclockwise direction and make the backside of switch touch E1 or E2, respectively. As a result, a conductive path is created for transmitting the pulse signals, as shown in Fig. 1a. The rotation angle can be detected by reading the number of pulses from the contacted electrode when the rotor keeps rotating along the same direction. Once the direction is changed, the switch can deflect to another electrode immediately to record the rotation angle along the reversed direction.

Similarly, for the linear TBD sensor, the bending and returning of finger will lead to the pulling and pushing of the FEP strip, respectively. Hence, the shear force caused by this linear motion will twist the switch to deflect/rotate toward either E1 or E2 electrode for defining the motion direction. The pulse signals can determine the bending angles.

There are several advantages of the proposed TBD sensors compare to the reported works[35,57,58]. Most of the current bidirectional triboelectric rotation sensors are featured with a two-disks design, which is attributed to the nature of the sliding mode of TENG and the corresponding signals. There are radial oriented circular grating patterns on each disk, and at least two sets of these circular grating patterns are arranged from the inner region to the outer region. Those patterns have phase differences in the alignment and are separately connected to two output channels. Hence, the rotation direction can be realized by identifying the leading phase between two channels, and the rotation angle can be counted through the output peaks. However, this method brings additional complexity to the programming part for phase identification. The requirement of the phase difference also leads to the requirement of the extra spacing which may lower the resolution. Although, the solution of adding the additional set of grating patterns was presented to improve the resolution. This design increases the whole size and the channels of the sensor. Moreover, this disk-shaped design can only act as a rotation sensor by attaching on the surface of the joint part, which limits the potential of universal applications based on a generic design. In contrast, our TBD sensor only needs one set of the basic grating patterns which are preferred for the further improvement of the sensing resolution by reducing the width and the size of the switch. The ring-shaped design not only can be used for the rotation sensing at the joint position, but also can be applied to detect the twist motion by going through the middle of the ring. Furthermore, this bidirectional switch enables the monitoring of the linear motions as well, by simply flattening the grating patterns. In general, by redesigning the base of the grating patterns for a specific purpose, these switches integrated TBD sensors can be considered as a universal system for the multidimensional monitoring of various motions.

**Design of an exoskeleton arm**. To apply our proposed sensors for monitoring the motions of arm and finger, an exoskeleton system is developed for system integration and demonstration. To achieve the cost-effective and customizable fabrication of the exoskeleton with good compatibility to various users, the entire single-arm exoskeleton consists of five 3D printed components, as illustrated in Fig. 2a, including a back supporter, an L-shaped shoulder module, an upper arm, a forearm, and a glove (also see Supplementary Fig. 1). The back supporter with an extended circular platform is fixed on the vest by screws, and the shaft of the rotational TBD back (RTBD-B) sensor is then installed on the platform for detecting the side raise motion of the upper arm. The fly ring part of RTBD-B is placed on one side of the L-shaped shoulder module. In addition, the shaft of the rotational TBD upper arm (RTBD-U) sensor is attached on another side of this shoulder module for sensing the forward and backward motion of the upper arm, and its fly ring is fixed on one side of the upper arm. For the shoulder joint of human, the monitoring of two DOFs motions can then be achieved by RTBD-B and RTBD-U. Next, the shaft of the rotational TBD forearm (RTBD-F) sensor is installed on another side of the upper arm, and the fly ring is then fixed on the forearm. As the human elbow can only possess one DOF, the RTBD-F is then in charge of detecting the corresponding motions.

Moreover, the human arm is also capable of performing the twisting actions, especially for the forearm, this action is essential to various tasks, such as the operations of tools. To capture the twisting motion, a groove is designed at the middle of the forearm part for fixing the shaft of the rotational TBD wrist (RTBD-W) sensor perpendicularly to allow the human arm to pass through.

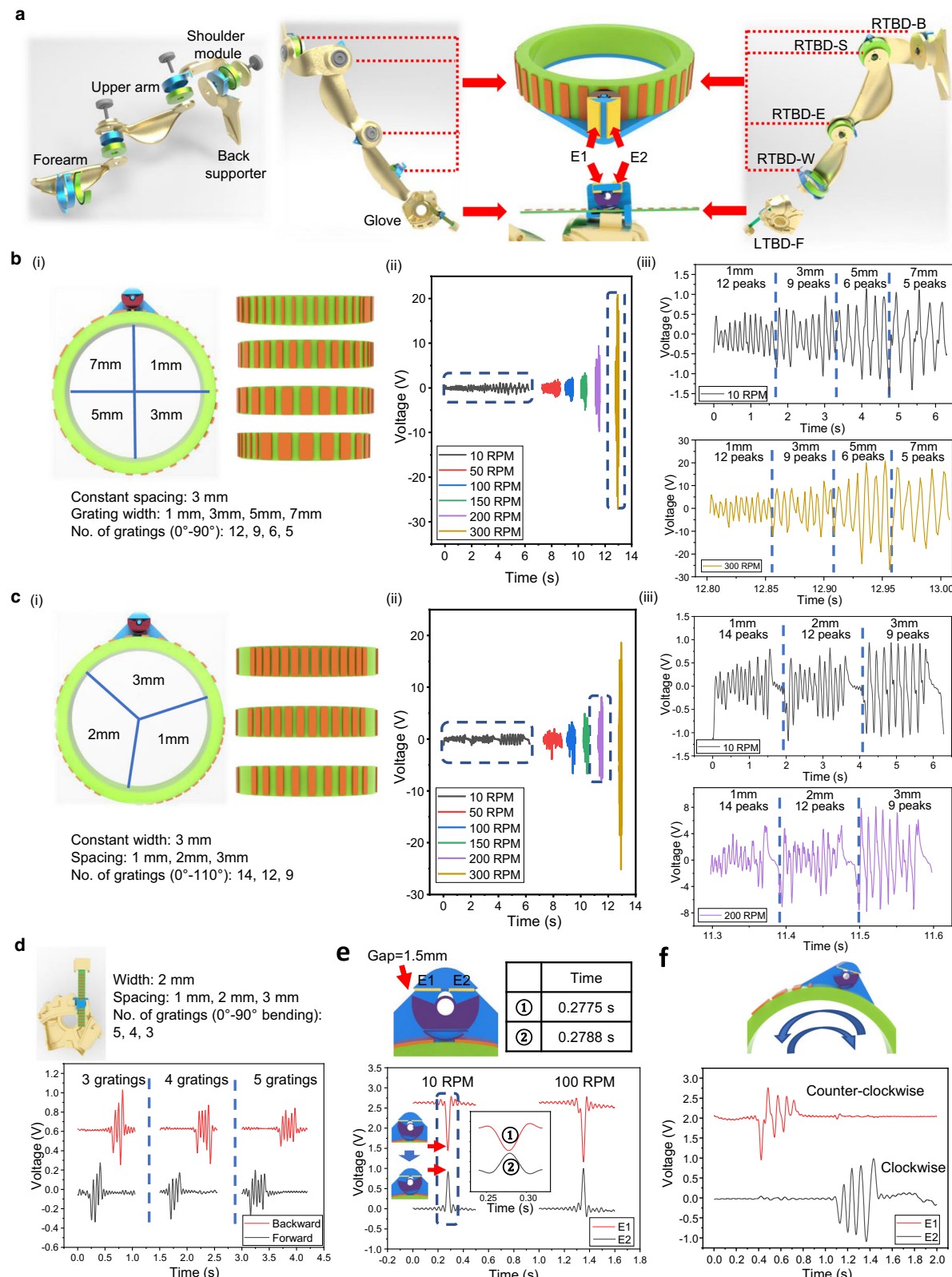

The fly ring is then modified with an ergonomic U-shaped structure in order to lock on the human arm gently. To further assemble all of four parts, including a back supporter, an L-shaped shoulder module, an upper arm, and a forearm, three bearing screws are applied on the joints of shoulder and elbow positions to reinforce the structure and maintain the smooth rotations. In addition, a glove has a palm case with fixtures and the separated finger cases. The linear TBD finger (LTBD-F) sensor is then attached on the palm case. A thick FEP based strip with the same grating pattern of PTFE is fixed on the finger case, and another side of the strip is inserted into the LTBD-F sensor on the palm case. The glove can achieve the detection of all five fingers by cascading the number of sensors depending on the applications.

**Fig. 2 Characterization and optimization of triboelectric bidirectional (TBD) sensors. a** Configuration of the assembly of the exoskeleton sensory system, with the rotational triboelectric bidirectional back (RTBD-B) sensor, the rotational triboelectric bidirectional shoulder (RTBD-S) sensor, the rotational triboelectric bidirectional elbow (RTBD-E) sensor, the rotational triboelectric bidirectional wrist (RTBD-W) sensor, and the linear triboelectric bidirectional finger (LTBD-F) sensor. **b** (i) Configuration of the varied grating widths (1, 3, 5, and 7 mm) with a constant spacing of 3 mm for a rotation TBD (RTBD) sensor, (ii) measurement of triboelectric output signals from the rotation speed of 10 revolutions per minute (RPM) to 300 RPM, and (iii) the enlarged waveforms of 10 RPM and 300 RPM. **c** (i) Configuration of the varied spacing (1, 2, and 3 mm) with a constant width of 3 mm for a RTBD sensor, (ii) measurement of triboelectric output signals from the rotation speed of 10 to 300 RPM, and (iii) the enlarged waveforms of 10 RPM and 200 RPM. **d** Configuration and measured triboelectric output signals for a linear TBD (LTBD) sensor with the varied spacing during the finger bending of 90°. **e** Response latency of switch (1.5 mm gap) during the changing of the rotation directions. The inserted graph is the enlarged waveform of 10 RPM, and the time of the peak voltages of the separation (1) and contact (2) signals are provided. **f** Measured signals for bidirectional rotation with the grating pattern of four varied widths (1, 3, 5, and 7 mm).

**Optimization and characterization of rotational triboelectric bidirectional (RTBD) sensor**. The signal intensity and the angular resolution of the RTBD sensor are two main factors that required optimization and characterization. As mentioned earlier, the contact area affects the intensity of triboelectric output, we conducted the tests by varying the width of PTFE strips as 1, 3, 5, and 7 mm, as shown in Fig. 2b(i). To ensure the consistency of the test condition, the area of the fly ring is equally arranged for attaching four types of strips on it with a fixed spacing of 3 mm. Specifically, there are 12 gratings for 1 mm width, 9 gratings for 3 mm width, 6 gratings for 5 mm width, and 5 gratings for 7 mm width. Additionally, in order to investigate the relationship between the signal intensity and the rotation speed, as well as the angular resolution at high rotation speed, the whole sensor is installed on a step motor for adjusting the rotation. The sensor was then tested for six different speeds: 10, 50, 100, 150, 200, and 300 RPM (revolutions per minute). In Fig. 2b(ii), the signal intensity increases as the rotation speed increases, and shows a significant enhancement after it reaches 200 RPM. This kind of increasing output may attribute to the shorter time of charge transfer. As the total amount of charges represented by the integration of the area of the single voltage peak remains the same, the narrower peak (shorter time of sliding across the single grating) will lead to higher peak amplitude. The enlarged output signals of 10 and 300 RPM are provided in Fig. 2b(iii). For the rotation speed of 10 RPM, all of four types of gratings are showing the correct number of peaks corresponding to the respective number of gratings. The averaged peak output voltage rises from 0.4 to 1 V, as the width of PTFE strip increases. For the rotation speed of 300 RPM, the numbers of gratings can still be clearly observed through those output peaks. Moreover, the averaged peak output voltage increases up to 18 V for the gratings of 5 and 7 mm width. Consider the fabrication precision of 3D printed rotational parts, the 3 mm width PTFE strip is selected for the grating pattern, in order to balance the resolution and the signal intensity under the potential noisy operating condition.

Secondly, for the selected 3 mm width strip, further investigation of the influences from the spacing distances are also conducted. The basic rule of assigning the spacing distance is to enable the generation of the differentiable peaks for two adjacent gratings when the copper spring of the switch is sliding across them. Hence, the contact width of the copper spring and the spacing are crucial to avoid the overlap of those signal peaks. According to the selected dimension of copper spring, the spacing distances of the grating pattern are choosing as 1, 2, and 3 mm, which divide the fly ring into three regions with equal areas (Fig. 2c(i)). Specifically, there are 14 gratings for 1 mm spacing, 12 gratings for 2 mm spacing, and 9 gratings for 3 mm spacing. Similarly, the sensor was then tested for six different speeds to evaluate the effect of the rotation speed on output signals: 10, 50, 100, 150, 200, and 300 RPM. As illustrated in Fig. 2c(ii), similarly,

the signal intensity increases as the rotation speed increases, and shows a significant enhancement after it reaches 200 RPM. On the other hand, although the widths of the gratings are kept the same, the enlarged output signal of 10 RPM shows that the spacing distance can also affect the signal intensity (Fig. 2c(iii)), which is increased from 0.5 to 0.8 V as the averaged peak voltage. The possible reason can be explained as, the narrower spacing will reduce the time of no contact between the gratings and the copper spring at a constant speed, and hence, the time of transferring the charges is reduced as well. As a result, the incomplete charge transfer process will eventually lower the peak output voltage for each grating. This phenomenon can be observed clearly at a higher speed, i.e., 200 RPM. In general, all the expected peaks for three types of spacings can be distinguished from 10 to 200 RPM. However, for 300 RPM, the 1 mm spacing gratings are showing the loss of the output peaks (see Supplementary Fig. 3). This issue can be addressed to the high rotation speed induced incomplete charge transfer process. In another word, the high speed and the narrow spacing cause the copper spring sliding almost continuously on those PTFE gratings, and hence, the output peaks will vanish occasionally. The possible solution of solving this issue is to redesign the copper spring with a smaller dimension to reduce the contact area against the gratings, so that the spring can still sense the narrower spacing at high speed. Based on the characterization tests for rotation sensing, the experimental results prove the feasibility of achieving the reliable detection ranging from low to high rotation speed.

In general, with the grating parameters of 1 mm width and 1 mm spacing, the angular resolution can achieve 4° for a sensor of 8.5 cm diameter. In terms of the applications of the wearable exoskeleton-based sensor, the human joints, i.e., the elbow, are usually moving at a rotation speed much lower than 300 RPM[59]. Hence, the proposed sensor-integrated exoskeleton is eligible for sensing the human motions with a reasonable resolution achieved by the cost-effective fabrication process, and projecting the corresponding motions into the virtual space or the robots to realize the intuitive, multidirectional, and quantitative manipulations. In addition, to further improve the sensing resolution for specific application which requires high precision, various fabrication approaches reported in the relevant researches can be adopted to increase the density of the gratings with the dimension of micrometer level, while maintaining the signal quality. MEMS process, screen printing, etc., have been demonstrated to fabricate the fine-featured gratings for TENGs[35,57,58,60,61].

Furthermore, to verify the influence of long-term usage on the signal quality, the reliability test is also conducted, as shown in Supplementary Fig. 4. A test sensor with 18 gratings was fabricated. The test data of 3 h was recorded at a rotation speed of 100 RPM. There is no significant decrease of the signal intensity, and all of 18 peaks above the threshold voltage are still clearly representing the

18 gratings. Thus, this reliability test can prove the robustness of the proposed sensor. The effects of humidity and temperatures are also evaluated as shown in Supplementary Fig. 5. Owing to the sensing strategy of peak counting with a proper threshold voltage, the decayed signal intensity at 95% relative humidity (RH) is still able to meet the sensing requirements. Additionally, this strategy also benefits the long-term functionality of the sensing signals during continuous operation.

**Optimization and characterization of linear triboelectric bidirectional (LTBD) sensor**. The LTBD sensor is applied for detecting the finger bending motion by converting the bending motions into the linear stretching of the sensor at the switch position. In Fig. 2d, the width of PTFE strip is reduced to 2 mm for the grating pattern by considering the limitation of the bending-induced displacement. The spacing distances of three different sensors are varied as 1, 2, and 3 mm. Hence, for a 90° bending, there are 5 peaks, 4 peaks, and 3 peaks detected for both forward and backward directions, respectively. The averaged peak voltage is about 0.2 V.

**Response latency of the switch for bidirectional sensing**. As an angular/rotational sensor, the capability of distinguishing the clockwise and counterclockwise rotations is a basic requirement that need to be investigated. As shown in Fig. 2e, a pendulum-shaped switch is integrated on the shaft base with two electrodes located at the backside. The shear force generated from rotation or linear stretching will lead to the deviation of the switch to the same side (opposite to the rotation direction). At the neutral state, two gaps between the switch and two electrodes are created to avoid the confusion of the generated signal. The functional signals can be sent out once the edge of the switch is connected with either one of the electrodes at the deviated state. However, the existence of these gaps may lead to the delay of response time during the changing of the direction. To investigate this issue, the PTFE strips are attached on both edges of the switch to make two simple contact-separation mode TENGs with the E1 and E2 electrodes. Hence, the contact and separation against E1 and E2 caused by the switching of the rotation direction will then generate the triboelectric outputs as the indicators. The total time required for the switch to rotate from E1 to E2 (or E2 to E1) can be obtained by observing the time difference between the peaks of the contact and the separation outputs, especially for the slow rotation speed. As a comparison, two different gaps of 1.5 and 3 mm are designed by tuning the shapes of two corners (see the data of 3 mm gap in Supplementary Fig. 6). In Fig. 2e of the data of 1.5 mm gap, the negative peak refers to the separation from the electrode, and the positive peak indicates the contact to the electrode. The experimental results of 10 and 100 RPM rotations are provided. Based on the enlarged output signals, the time period between separation and contact of the switch is about a few milliseconds, which is the time required for changing the sensing direction back or forth. Therefore, by considering the general condition of the human motion speed, there is almost no significant time delay during the switching, even at a slow rotation speed of 10 RPM. Moreover, the results of both 1.5 and 3 mm gaps prove this design possesses a good response time for bidirectional sensing, and these gaps can be further reduced by simply changing the edge design of the switch, e.g., a sharper angle of the edge. A test result of bidirectional rotation sensing is also provided in Fig. 2f, four gratings with different widths are selected, including 1, 3, 5, and 7 mm. It shows the general function of the RTBD sensor.

**Signal processing for real-time manipulation**. Wearable triboelectric sensors are usually encountered with signal fluctuations caused by the body motions and the wiring connections. Especially for the continuous and digitized sensing of the rotation angles, the consistency of the signals is necessary for enabling the accurate recognition of the corresponding outputs. The variations of peak voltages will bring difficulty in the programming of the peak recognition. Hence, an external circuit consists of the operation amplifier and the comparator is developed for the microprocessor of Arduino, as shown in Fig. 3a. The threshold voltages are tunable by the resistance in the comparator circuit. As a guideline, the threshold voltage will be set as low as possible for recognizing the weak signals during the slow motions, but also need to be much higher than the background noise to avoid the false detection. After the preprocessing circuit, the original triboelectric waveforms will be converted into the square waveforms for counting the effective peaks, as illustrated in Fig. 3b (also see Supplementary Fig. 7).

With the assistance of this approach, the triboelectric sensory information can be easily applied into the programming process. For each sensor, there are two channels in charge of bidirectional sensing. The output peaks of two channels can be programmed to print the specific numbers, i.e., 1 for forward rotation, and 2 for backward rotation (see Supplementary Movie 1). For the grating pattern with 3 mm width of PTFE strip and 3 mm spacing, each grating represents an angle of 10°. Hence, if there are five 1 (e.g., 11111) printed out during rotation, the sensor is rotated forward by 50°. For four 2 (e.g., 2222), the sensor is rotated backward by 40°.

**Demonstration of manipulation in virtual space**. Considering the drastic advancement of VR/AR technologies, different types of training or entertainment software are developed to enrich the user experiences. In terms of HMI hardware, the current devices are mainly the hand-held controllers to capture the hand motions and the spatial position. The button-based interaction is still lack of intuitiveness. Although there are several companies presenting the data glove with inertial or resistive sensors for monitoring the finger activities. A universal solution for projecting the motions of the whole arm with low power consumption is necessary to pave the way for effective and long-term usage of those training or entertainment software. Hence, the primary demonstration of projecting the human arm motions into the virtual character was conducted at first (see Supplementary Movie 2). As depicted in Fig. 3c(i), the controllable motion ranges of virtual character are labeled according to the DOFs of exoskeleton arm (not human arm), including 120° of side raising, 180° of forward lifting, 200° of elbow bending, 270° of wrist twisting, and 90° of finger bending.

The programming part includes signal readout code in Arduino, signal processing and visualization code in Python, motion control code in Unity. The serial communication code between Arduino and Python, as well as Python and Unity, are also required. For the real-time control, the number-based commands are generated by the detection of the peaks from specific channels/joints. Next, the motion control code in Unity will link those numbers to the respective joints on virtual character, i.e., shoulder, elbow, wrist, etc. In Fig. 3c(ii), by setting 10° per peak as the rotation rate of virtual joints, the corresponding signals for controlling the virtual character to reach the final posture are illustrated. Noticeably, the rotation rate can also be tuned in motion control code to achieve the different projection ratio of arm motions, i.e., 30° of virtual elbow rotation by rotating the real arm 10°.

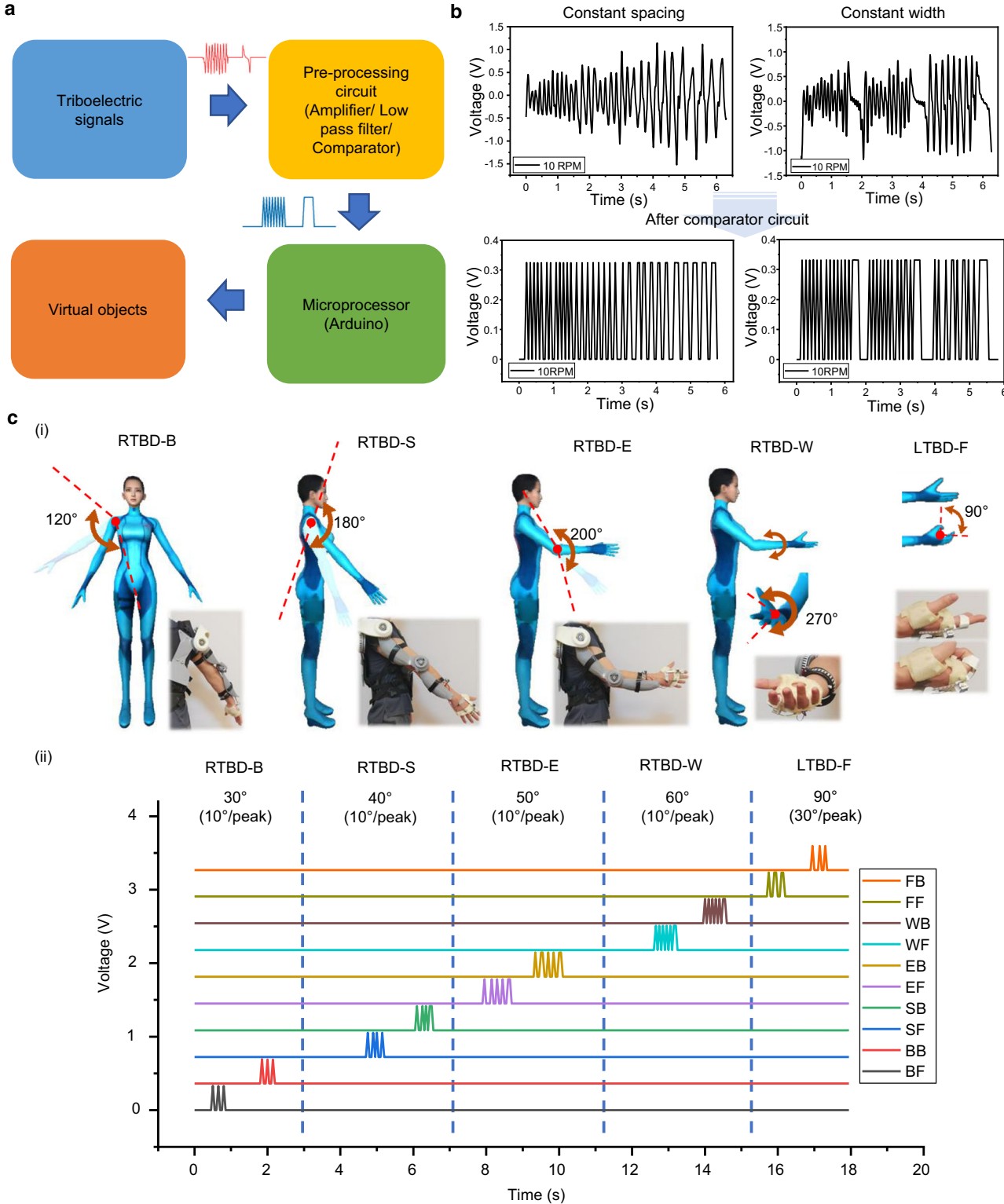

**Fig. 3 Signal processing and demonstration in virtual space. a** Flow chart of triboelectric signal processing for manipulation in virtual space. **b** Examples of the original triboelectric signals after the preprocessing circuit. **c** Demonstration of controlling virtual character: (i) the controllable motions and the activated sensors with the limits of motion ranges for the rotational triboelectric bidirectional back (RTBD-B) sensor, the rotational triboelectric bidirectional shoulder (RTBD-S) sensor, the rotational triboelectric bidirectional elbow (RTBD-E) sensor, the rotational triboelectric bidirectional wrist (RTBD-W) sensor, and the linear triboelectric bidirectional finger (LTBD-F) sensor and (ii) the real-time signals during the manipulation, the output channels of BF, BB, SF, SB, EF, EB, WF, WB, FF, and FB stand for the forward (clockwise) and the backward (counterclockwise) rotations of the RTBD-B sensor, the RTBD-S sensor, the RTBD-E sensor, the RTBD-W sensor, and the LTBD-F sensor, i.e., BF for forward rotation of RTBD-B. Photo credit: Minglu Zhu, National University of Singapore.

**Demonstration of robotic arms control**. Robotic manipulation is essential for industrial production, medical operation, and daily assistance. Currently, the conventional techniques include the joystick, touchpad, and wearable devices based on inertial or resistive sensors. However, to achieve the efficient parallel control in real-time, the low power consumption and highly customizable sensory system is still required for further researches to satisfy the requirements in the industrial automation, rehabilitation, and training program in cyberspace. The proposed exoskeleton arms with the TBD sensors are then applied to realize the intuitive manipulation of robotic arms for completing a specific task.

A human-like robotic arm consists of five motors which are operated by the motor controller, including two motors on the shoulder for two DOFs motions, one motor on the elbow, one motor on the wrist, and one motor for the gripper. In Fig. 4a, the command with the format of hexadecimal is accepted for communicating with the controller. Hence, the previous command of decimal number will be converted into hexadecimal at first before sending to the controller. Based on the Supplementary Movie 3, for checking the functionalities of the exoskeleton arm, the multidirectional and multi-degree control was performed to prove the feasibility.

Next, a comprehensive demonstration of conducting the dexterous manipulation of two robotic arms was achieved. The entire task can be divided into several steps, starting from the movement of the black empty box, to the dropping of the grabbed cube into the box. The detailed motion and the corresponding output signals from each sensor are provided as shown in Fig. 4b(i), (ii). Two microprocessors are used to record the signals from two exoskeleton arms. Firstly, the left arm is lifted up 20° as the side raising via the detection of RTBD-B sensor (step 1). A forward lifting of 60° is then performed by sensing the rotation from RTBD-S sensor (step 2). Afterward, the forearm is moved toward body with 70° through the signals from RTBD-E sensor at the elbow (step 3). Lastly, the hand is twisted backward (counterclockwise) by 90° to adjust the orientation of the black box through RTBD-W sensor at the wrist (step 4). For the right arm, it conducts a side raising by 30° through the signals from RTBD-B sensor, and move to the cube holder (step 5). Then, the finger is bent 90° inward to grab the cube by monitoring the signals from LTBD-F (step 6). Similarly, the arm is then lifted forward and moved toward the body by 60° and 70°, respectively (step 7, 8). Next, the hand is twisted forward (clockwise) by 70° to adjust the orientation of the cube (step 9). Finally, the finger is opened 60° to drop the cube into the box (step 10). The reason for a larger bending angle of 90° during the grabbing of the cube is to increase the force to grab it firmly, and the returning of 60° is enough to release the cube.

This integrated robotic demonstration as a primary result, proves the feasibility of utilizing the low-cost, energy-saving sensors to achieve the dexterous manipulation of robots, which can be further scaled up for realizing the parallel control of multiple robots in real industrial applications. Hence, this approach offers an easier way to accomplish the reprogramming jobs for the robotic arms.

**Demonstration of ping-pong game**. Owing to the quantitative detection of the multidimensional motions for all of the arm joints, the virtual interactions are able to be performed more accurately. The projection of the entire motion chain of the arm under a complex task can greatly improve the effectiveness of the training program, due to the better consistency between the real and the virtual activities. Especially for sports and rehabilitation program, although these applications show great potentials in terms of the efficiency in performance improvement, the frequent motions and the requirements of special movements become a challenge to the sensing system. Hence, a demonstration of ping-pong game was presented to verify the integrated manipulation in completing the particular task (see Supplementary Movie 4). More specifically, this demonstration is designed to be a training program for monitoring the joint motions during a specific strike (Fig. 5a). Hence, the Python code will record the complete strike action of real player for strike recognition, and then, the corresponding command of strike action will be sent to Unity program. There are four strike actions performed in this program, including forehand stroke, right sidespin, left sidespin, and smash. Unlike the synchronized manipulation shown in Fig. 3b, the control command can be triggered only if the correct joint motions are detected by the sensors. As shown in Fig. 5b, for a forehand stroke, both RTBD-S sensor and RTBD-E sensor were rotated forward as the shoulder and the elbow were lifted up to strike the ball. The corresponding signals are shown in Fig. 5c. Similarly, for left or right sidespin, RTBD-B sensor and RTBD-W sensor were activated as the shoulder was side raised and the wrist was twisted accordingly. For a smash, there were mainly three sensors engaged during a complete strike path, including the RTBD-B, RTBD-E, and RTBD-W. The respective signal peaks of each sensor for each strike can indicate the motion status of the real human arm, and hence, to monitor whether the arm is following the correct trajectory. This approach plays a key role for either training the beginners with action correction capabilities in virtual space or monitoring the mobility of disabled patients in the rehabilitation process

**Force estimation with kinetic analysis of sensory information**. To enhance the consistency and intelligence of the motion projection between human and the HMI manipulated objects, the collection of diverse physical parameters is necessary. For instance, in terms of those humanoid robotics, the operations with accurate displacement and speed of linear and rotational motions are highly desirable. Meanwhile, for the sports training or healthcare program in virtual space, the input information of acceleration and force are also crucial to improve the immersive experience for better training outcomes. Currently, two basic approaches are adopted to accomplish those tasks. One of them is to introduce multiple sensors which are in charge of the detection of different mechanical stimulus, such as force, strain, displacement, inertial, etc. However, these various sensors will definitely increase the system complexity and the power consumption for long-term sustainability[62]. As an alternative, another method is to implement the kinetic analysis of the existed sensors for extracting more dynamic information other than the primary sensing signals. As an example, the inertial sensor is frequently used for detecting acceleration and attitude, but it also can be utilized to obtain the impact force via a proper algorithm. Hence, the strategy of applying the kinetic analysis to realize the multifunctional sensing with a single type of sensor can be considered as a promising solution which will bring more convenience to the system integration and the processing of the sensing signals.

For the proposed RTBD sensors, the rotation detection relies on the pulse signals as mentioned before. As the spacing of two adjacent PTFE gratings represents 10° of angular rotation, the time interval between two pulses can indicate the time required for rotating 10°. Therefore, the instantaneous average velocity can be calculated directly as below:

$$\text{RPM} = \frac{60}{t_p * N} \tag{1}$$

Where $t_p$ is the time interval between two pulses, $N$ is the number of the gratings of the fly ring.

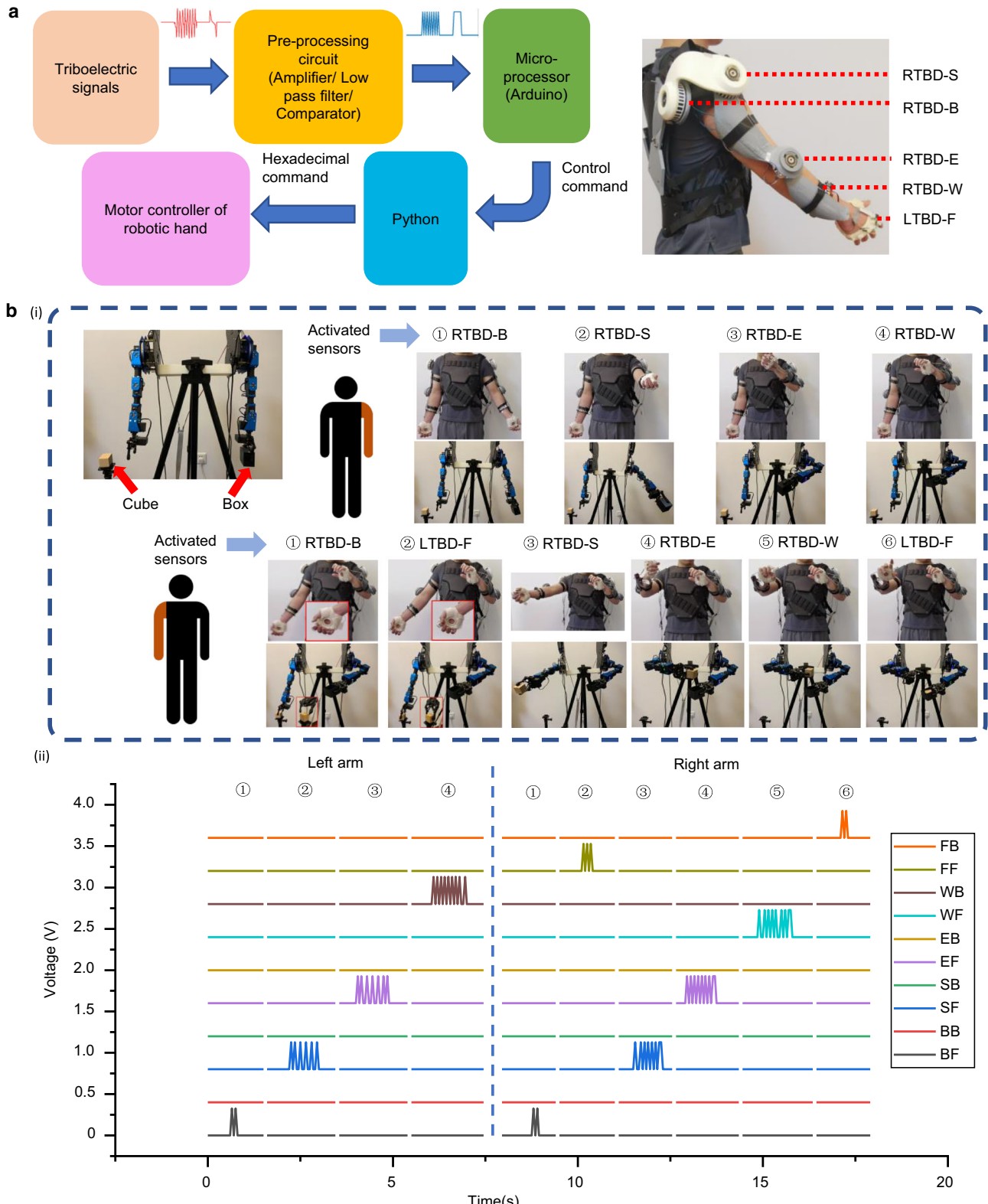

**Fig. 4 Signal processing and demonstration in robotic control. a** Flow chart of triboelectric signal processing for manipulation of robotic arms, and the photograph of the exoskeleton sensory system (right arm) with the rotational triboelectric bidirectional back (RTBD-B) sensor, the rotational triboelectric bidirectional shoulder (RTBD-S) sensor, the rotational triboelectric bidirectional elbow (RTBD-E) sensor, the rotational triboelectric bidirectional wrist (RTBD-W) sensor, and the linear triboelectric bidirectional finger (LTBD-F) sensor. **b** Demonstration of joint works of two robotic arms for picking up the cube and placing it into the box, (i) flow chart of the motions, and the activated arm and the sensors. (ii) the real-time signals during the manipulation, the channels of BF, BB, SF, SB, EF, EB, WF, WB, FF, and FB represent the forward (clockwise) and the backward (counterclockwise) rotations of the RTBD-B sensor, the RTBD-S sensor, the RTBD-E sensor, the RTBD-W sensor, and the LTBD-F sensor. Photo credit: Minglu Zhu, National University of Singapore.

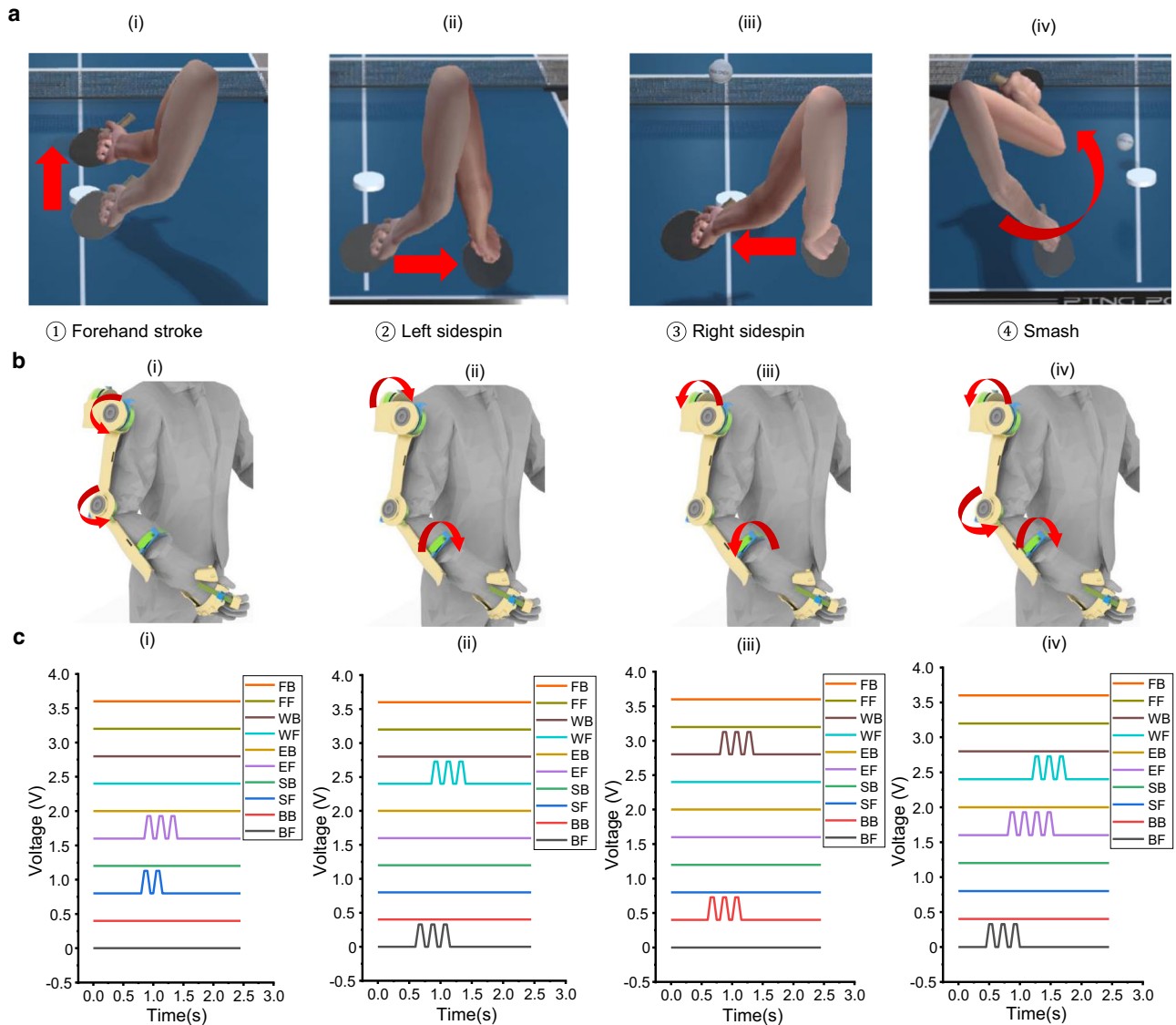

**Fig. 5 Demonstration of ping-pong training program for simultaneous monitoring of multiple sensing signals. a** Illustrations of four striking motions in ping-pong game, (i) forehand stroke, (ii) left sidespin, (iii) right sidespin, and (iv) smash. **b** The main activated sensors on exoskeleton for the corresponding strikes, the rotation direction of each sensor is marked by red arrow. **c** The real-time signals generated from four strikes, the channels of BF, BB, SF, SB, EF, EB, WF, WB, FF, and FB represent the forward (clockwise) and the backward (counterclockwise) rotations of the rotational triboelectric bidirectional back (RTBD-B) sensor, the rotational triboelectric bidirectional shoulder (RTBD-S) sensor, the rotational triboelectric bidirectional elbow (RTBD-E) sensor, the rotational triboelectric bidirectional wrist (RTBD-W) sensor, and the linear triboelectric bidirectional finger (LTBD-F) sensor.

By integrating this simple calculation into the Python code, the rotation velocity sensor can be realized. In the meantime, owing to the specialized design of the exoskeleton arm, the position of the rotation TBD sensors and the distance between those sensors are in good accordance with the position of the human joints and the length of the human arms, respectively. These are the advantages for utilizing the sensory information to achieve the advanced kinetic analysis of human motions. As a verification, the straight punch/jab in boxing was conducted for preliminary investigation, as shown in Fig. 6a(i). To simplify the analysis, the shoulder joint is considered as a fixed point by assuming no torsional motion, and the entire motion paths of three joints, where $L_1$ is the length of the upper arm, $L_2$ is the length of the forearm, α is the angle between the upper arm and the centerline of punching, β is the angle between forearm and the centerline of punching, and γ is the angle between the upper

arm and the forearm. In Fig. 6a(ii), the output signals of RTBD-S and RTBD-E sensors represent the data obtained from the initial state (40° of γ and 90° of α) till the final state during a complete punch. As a result, for the same displacement of the fist, there are 14 peaks and 9 peaks for RTBD-E sensor and RTBD-S sensor, respectively. Therefore, the RTBD-E sensor has a better resolution for this activity.

Figure 6b(i) illustrates the schematics of the entire kinetic analysis of the straight punch. The summation of α, β, and γ always equals to 180° according to the basic triangular theorem. Moreover, as both the shoulder and the fist are fixed on the centerline, there is a predictable relationship among three angles for a specific position of the fist. The relationships of three angles along the entire punching path are presented in Fig. 6b(ii) through the actual measurements (see Supplementary Table 1). In general, this movement can be treated as a slider-crank

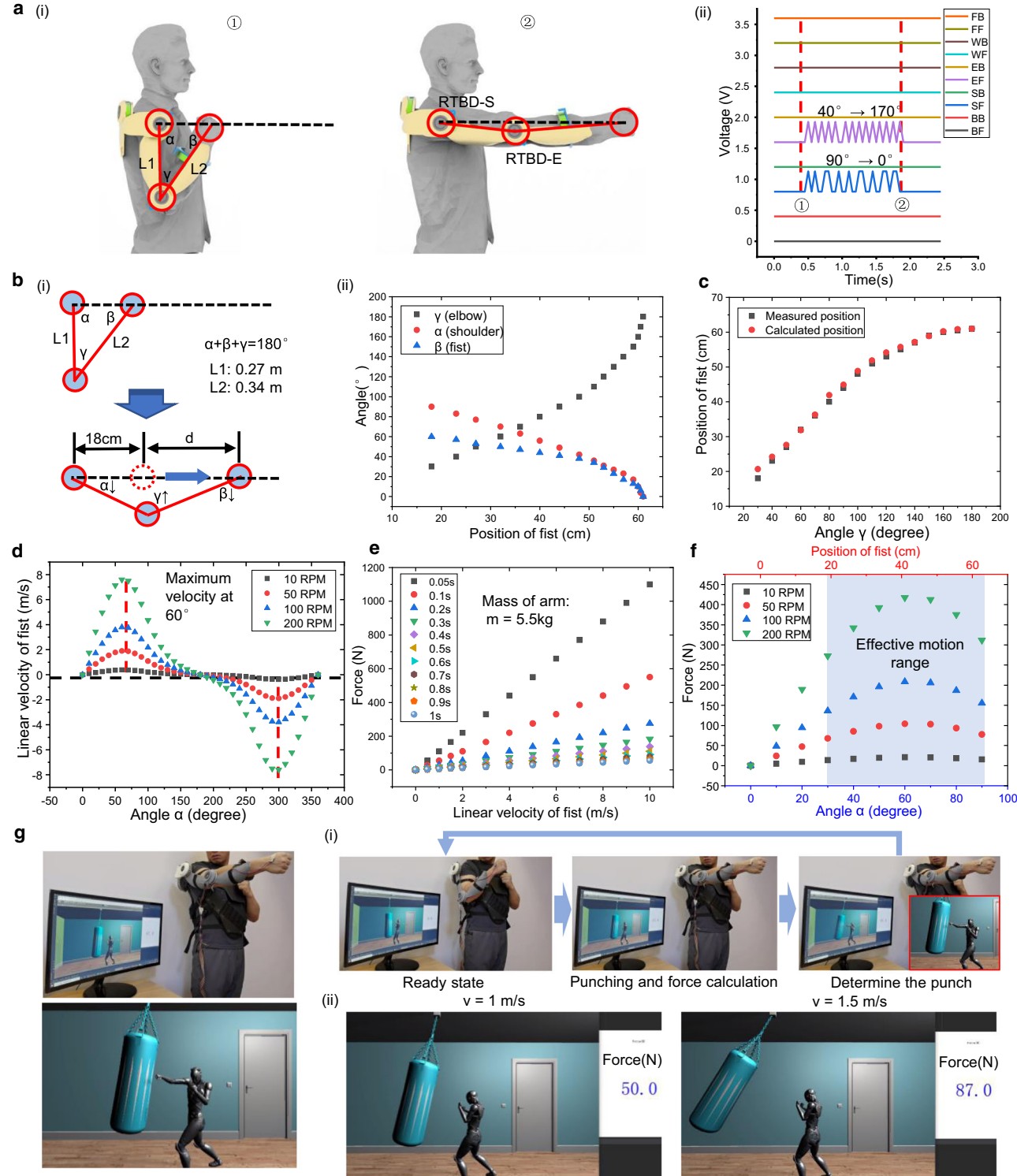

mechanism for further study[63]. The length between the shoulder and the fist can be expressed as:

$$L = L_1 \cos\alpha + L_2 \cos\beta \qquad (2)$$

Hence, let $\lambda = L_1/L_2$, the position of the fist can be calculated as:

$$d = L_{\max} - \left( L_1 \cos\alpha + L_2 \left( \sqrt{1 - (\sin\alpha)^2 \lambda^2} \right) \right) \qquad (3)$$

Where $L_{\max}$ is the maximum distance between the shoulder

and the fist. However, as mentioned before, the resolution of RTBD-S sensor for $\alpha$ angle is lower than RTBD-E sensor for $\gamma$. Owing to the fixed relationship of three angles, the $\alpha$ angle can then be determined by RTBD-E sensor. As shown in Fig. 6c, the calculated positions of the fist via the sensing data show the good consistency with the measured positions of the fist.

Furthermore, the velocity of the fist can be determined as:

$$v = \frac{dd}{dt} = \frac{dd}{d\alpha} \cdot \frac{d\alpha}{dt} \qquad (4)$$

**Fig. 6 Force estimation with kinetic analysis of sensory information from rotational triboelectric bidirectional (RTBD) sensor. a** (i) Schematics and (ii) the generated signals from the rotational triboelectric bidirectional shoulder (RTBD-S) sensor and the rotational triboelectric bidirectional elbow (RTBD-E) sensor during the punch, the channels of BF, BB, SF, SB, EF, EB, WF, WB, FF, and FB represent the forward (clockwise) and the backward (counterclockwise) rotations of the rotational triboelectric bidirectional back (RTBD-B) sensor, the rotational triboelectric bidirectional shoulder (RTBD-S) sensor, the rotational triboelectric bidirectional elbow (RTBD-E) sensor, the rotational triboelectric bidirectional wrist (RTBD-W) sensor, and the linear triboelectric bidirectional finger (LTBD-F) sensor. L1 and L2 represent the length of the upper arm and the forearm, α, β, and γ represent the angles between the upper arm and the centerline, the forearm and the centerline, as well as the forearm and the upper arm. **b** (i) Schematics and (ii) measured data for the relationship between three angles and the position of the fist. **c** Comparison between the measured position and the position calculated by the sensory information. **d** The estimated instantaneous linear velocity of the fist against the varied angle of α for the given rotation speeds of 10, 50, 100 and 200 revolutions per minute (RPM). **e** The estimated punching force exerted on the target against the changing velocity of the fist and the variation of the duration for stopping, the mass of the participant's arm is ~5.5 kg. **f** The estimated punching force exerted on the target against the varied angle of α for a specific rotation speed, the actual effective motion range of the fist is shaded. **g** Demonstration of punching force estimation through the data of the rotation sensing from the RTBD sensors in virtual space, (i) process of determining the level of the virtual punch based on the rotation sensing information, (ii) light punch and heavy punch demonstrations. Photo credit: Minglu Zhu, National University of Singapore.

Since the angular velocity of RTBD-S sensor can be measured as $\omega_S = \frac{d\alpha}{dt}$, the velocity of the fist can then be calculated as:

$$v = -L_1\omega_S\sin\alpha - L_2\omega_S\frac{\lambda^2\sin\alpha\cos\alpha}{\sqrt{1 - \lambda^2(\sin\alpha)^2}} \qquad (5)$$

By considering the circular rotation of RTBD-S sensor and plotting the graph for the linear velocity of the fist against α angle shown in Fig. 6d, the maximum linear velocity can be observed at the α angle of ~60°. For the fast punch with 200 RPM, the velocity of ~7.6 m/s is calculated, which shows a good agreement according to the literature data. Additionally, in terms of data interpretation for other motions, the linear acceleration can also be obtained through the second derivative of the linear displacement function. Next, the punching force can also be defined through the momentum conservation equation:

$$mv - mv_0 = F_t t_c \qquad (6)$$

Where $m$ is the mass of the arm, $v_0$ is the initial velocity of the fist, $v$ is the final velocity of the fist, $F_t$ is the force received by the target, $t_c$ is the time period from the contacting of the target to the complete stop of the fist. Since the $v_0 = 0$ m/s, $F_t = \frac{mv}{t_c}$. Hence, the punching forces for different contact periods and different linear velocities are shown in Fig. 6e. In Fig. 6f, for a contact period of 0.1 s, the punching forces against the α angle are plotted under different angular velocities, and the actual effective motion range is shaded in blue.

As a practical demonstration based on the previous kinetic analysis, the boxing training program in Unity was designed and performed with the exoskeleton arm (see Supplementary Movie 5). After programming the relevant equations into the Python code for processing the signals of the rotation angles, the corresponding command can be sent to the Unity character in response to the real arm motions. As shown in Fig. 6g(i), once the punching action is performed, the maximum rotation speed is then recorded for further calculation of the punching force. Afterward, the virtual punch with different forces will be initiated accordingly. Meanwhile, the estimated punching force is simultaneously displayed, a punch of 50 N force was obtained at the velocity of 1 m/s, and a punch with 1.5 m/s generated 87 N force (Fig. 6g(ii)). In order to present the effect of different punching force in virtual space, a sandbag was added as the target with a physical effect. Therefore, a heavier punch will cause larger knock-off than a gentle punch. In order to evaluate the accuracy of the estimated force, a punch force meter was applied to test the actual force exerted by arm with the same velocity (see Supplementary Fig. 8a(iii)). The forces of 1 and 1.5 m/s punch

are measured as 4 and 6 kg, respectively, and are equivalent to 40 and 90 N. The errors may attribute to several aspects, such as the inconsistency of punching motions of human, loss of the decimal places of the force meter, and fluctuation of angular velocity due to the loss of the detection of the gratings during rotation. The sensor related issues can be solved by applying micro-machining techniques or MEMS process with higher precision for fabrication.

In general, without adding other types of sensors, the proper utilization of the sensing signals from the existed TBD sensors can effectively explore the capabilities of performing the multifunctional monitoring with minimal optimization (see Supplementary Fig. 9). By further developing the lower limb exoskeleton, after collecting the dimensions of the lower limbs (length, mass, etc.), the walking pace, the walking speed, the step length, and even the stepping force are able to be estimated. As a result, the system with the capability of full-body monitoring can conveniently track the status of activities for both the athletics and the patients in rehabilitation. Owing to the facile designed low-cost TBD sensors and the exoskeleton, the whole system does not only provide an economic and universal solution for capturing the complex human motions, but also introduces a strategy of expanding its functionalities via the specific analysis of the original sensing data.

## Discussion

In terms of HMIs for both real and virtual worlds, the accurate parallel control is essential to ensure the efficiency of conducting the complex tasks in various scenarios, ranging from industrial robotic automation to personal human–machine interaction. As an advanced wearable device which is still under the development, the exoskeleton is mainly serving as an assistive tool with various actuation mechanisms for improving the mechanical output power of human. Several designs were proposed, ranging from rigid metallic structure, to the soft exosuit featured with the fabric design and the wire-based actuator. In contrast, the researches of new body motion sensors with the platform of a exoskeleton for the HMI applications is rarely reported. The conventional solutions with the inertial and the resistive sensors suffers from several drawbacks, such as the large power consumption, and the huge amount of data for back-end processing. To promote the related HMI technologies for improving the real-time parallel control, the sensory system with low cost, low power consumption, and low data complexity is urged. Herein, we proposed a TBD sensor which can be universally applied on different joints of the exoskeleton arm for capturing and projecting the motions of the entire upper limbs. Differs from the

multilayers of special designed grating patterns adopted by other bidirectional triboelectric rotation sensors, this TBD sensor with a switch structure and a basic grating structure can realize the bidirectional sensing for both rotational and linear motions from different joints, including multiple DOFs rotations of the shoulder and twisting of the wrist. This single and facile design with multi-functionalities greatly reduce the complexity of fabrication and maintenance, and also simplify the back-end signal processing. In the meantime, the sensor shows a good performance from low speed of 10 RPM to high speed of 300 RPM, proves the feasibility for different application scenarios. Moreover, this design also offers excellent tunability of the size and the resolution of the sensor by altering the dimensions of the grating pattern and the switch with micro-machining approaches.

On the other hand, owing to the good consistency between exoskeleton design and the structure of the human body, a further kinetic analysis of the sensing information can provide more details of other physical parameters in addition to the original data of rotation angle, such as displacement, velocity, force, etc. Eventually, the functions of the inertial sensor, force sensor, vision sensing etc., can be partially replaced for a certain purpose, and this also reduces the complexity of the sensing system by applying the proposed sensor with the self-generated signals. Leveraging the fast establishment of 5 G network infrastructures, numerous intelligent robotics in real-world and programs in cyberspace are requiring the user-friendly, multifunctional, and affordable solutions of HMIs to satisfy various demands. With the features of low cost and power consumption, as well as less data complexity, the proposed TBD sensor enabled exoskeleton sensory system offers an economic and universal solution for realizing the real-time parallel control via the multidimensional body motion tracking, which may benefit the advancements in industrial automation, unmanned shop and warehouse, digital twin, rehabilitation, training program, etc.

## Methods

**Fabrication of exoskeleton arm and TBD sensor structures**. Exoskeleton arm and main structures of TBD sensors, including a back supporter, an L-shaped shoulder module, an upper arm, a forearm and a glove for exoskeleton arm, and shafts, fly rings, switches for TBD sensors, were designed by Solidworks 2018, and 3D printed by Anycubic 4 Max Pro using polylactic acid (PLA) filament.

**Fabrication of TBD sensor**. PTFE film was dice into the strips with designed width, including 1, 3, 5, and 7 mm. Lengths of strips are determined by the thickness of the 3D printed fly ring of sensor, varied from 10 to 20 mm for the proposed exoskeleton arm. All of the strips were bonded with Al electrodes by Ag conductive epoxy, and the strips were then attached on the fly ring based on the designed spacing, including 1, 2, and 3 mm, in order to form the grating patterns. An additional Al electrode was added as interconnector of all the PTFE gratings on the fly ring to make a comb structure for output enhancement. Similar procedures were conducted to make the PTFE grating patterns on flexible FEP strip of the linear TBD sensor.

An arch shaped copper spring was attached onto the 3D printed switch. Two Al electrodes were then connected to the spring at the opposite sides and extended to the backside of the switch. The switch was then installed onto the holder of the shaft at the corner with a rotatable screw. The stand of the holder was also attached with two Al electrodes at the opposite sides for making the readout channels when the switch contact with one of these electrodes.

**Assembly of exoskeleton arm and TBD sensors**. Platforms or fixtures for TBD sensors were designed on all of five exoskeleton components, including the back supporter, the L-shaped shoulder module, the upper arm, the forearm, and the glove. For each fabricated RTBD sensor, the shaft and the fly ring were separately installed on two connected exoskeleton components, i.e., the shaft of rotational TBD sensor of back (RTBD-B) was installed on the back supporter, and the fly ring of RTBD-B sensor was installed on the L-shaped shoulder module. Next, two connected exoskeleton components were assembled by using the bearing screw to fix the rotation center. The same procedures were repeated for shoulder (RTBD-S) and elbow (RTBD-E) sensors, and hence, the main structures of the entire arm were completely connected. For the wrist sensor (RTBD-W), a groove on the forearm exoskeleton was designed to install the sensor perpendicularly. The fly ring

of RTBD-W was cut for an opening for the ease of wearing by forearm. The inner surface of the fly ring was reshaped to two parallel surface, in order to lock on the wrist gently.

For linear TBD sensor of finger (LTBD-F), the switch holder was installed on the fixture of the 3D printed palm case of the glove. The flexible FEP strip was then inserted into the switch, and another end of the strip was fixed on the finger case.

**Characterization of triboelectric output**. For triboelectric output from TBD sensors, the voltages were measured by oscilloscope (Agilent, InfiniiVision, DSO-X 3034 A). A 42 stepper motor (17HS4401 NEMA) with an Arduino Uno based motor controller was used to provide the tunable rotations for testing the RTBD sensor from 10 to 300 RPM.

**Design of preprocessing circuit for microprocessor**. Arduino MEGA 2560 was used to process the signals for performing the robotic control and the virtual space interactions. A preprocessing circuit consists of LM358 operation amplifier and LT1017CN8 comparator was designed and attached on the Arduino microprocessor for the readout of the triboelectric signals from the proposed sensors.

**Demonstration of robotic control**. Two robotic arms (Hiwonder xArm 2.0) with the servo controllers were used for conducting the demonstration. The triboelectric signals processed by Arduino MEGA 2560 were delivered into the Python code for sending the hexadecimal commands to the servo controllers. A conversion port (CH340G) module was used to transmit the commands from the computer to the servo controller. The authors affirm that the participant provided informed consent for publication of the images in Figs. 1a, 3c, 4a, b, 6g, Supplementary Figs. 1c, 8a, and the videos in Supplementary Movie 2, Supplementary Movie 3, Supplementary Movie 4, and Supplementary Movie 5.

**Reporting Summary**. Further information on research design is available in the Nature Research Reporting Summary linked to this article.

## Data availability
The data that supports the findings of this study is available from the corresponding authors upon reasonable request.

## Code availability
The codes that support the findings of this study are available from the corresponding authors upon reasonable request.

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

## Acknowledgements
This work was supported by the National Research Foundation, Singapore under its AI Singapore Program (AISG Award No: AISG-GC-2019-002); the National Key Research and Development Program of China (Grant No. 2019YFB2004800, Project No. R-2020-S-002) at NUSRI, Suzhou, China; "Intelligent monitoring system based on smart wearable sensors and artificial technology for the treatment of adolescent idiopathic scoliosis", the "Smart sensors and artificial intelligence (AI) for health" seed grant (R-263-501-017-133) at NUS Institute for Health Innovation & Technology (NUS iHealthtech); the Collaborative Research Project under the SIMTech-NUS Joint Laboratory, "SIMTech-NUS Joint Lab on Large-area Flexible Hybrid Electronics"; and National Research Funding—Competitive Research Program (NRF-CRP) (R-719-000-001-281). Any opinions, findings and conclusions or recommendations expressed in this material are those of the author(s) and do not reflect the views of National Research Foundation, Singapore.

## Author contributions
M.Z., T.C., and C.L. conceived the idea. M.Z. and C.L. planned the experiments. M.Z. designed and completed the hardware, and performed the experiments. M.Z. took all the photos shown in figures. M.Z. and Z.S. wrote the control programs and algorithms for demonstration. M.Z., Z.S., and C.L. contributed to the data analysis and drafted the manuscript. M.Z. and C.L. edited the manuscript.

## Competing interests
M.Z., Z.S., and C.L. are inventors on patent application (pending, Ref: 2021-019) submitted by National University of Singapore, that covers exoskeleton manipulator with the bidirectional triboelectric sensors enabled sensory system. The remaining author declares no competing interests.
