## [Peer Review File · Nature Communications]

Reviewer #1 (Remarks to the Author):

The authors propose a customized exoskeleton system with triboelectric bi-directional (TBD) sensors to realize the detection of both linear and rotational motions from human regarding joints rotation/twisting and finger bending. The switch used in TBD sensor offers great convenience to simplify the design of the grating patterns and the reading of the generated signals for performing the quantitative monitoring of bi-directional motions. In contrast to the recent research trend in the flexible wearable sensors (e.g., *Nature* 587, 219–224 (2020)), this exoskeleton based sensory system possesses its own superiority in the accurate detection of various motions. As a result, with the proper data analysis and the demonstrations with high completeness, the authors proved the dexterous motion projections in both VR and robotic manipulations, and also shown the possibilities of obtaining other kinematic parameters. This proposed exoskeleton sensing system illustrates a universal solution with various functionalities, which can be beneficial in diversified HMI applications. I think this work is very in time and do very comprehensive investigations on low-cost interface for human-robotic parallel control. Hence, I recommend the minor revision with the following comments.

1. As the rotational sensors, the authors demonstrated the sensing resolution of about 10 degree, and can be optimized via the design of grating patterns. Can the authors comment on the possible limit of resolution at current stage? And how about the actual sizes of these sensors?
2. The whole sensory system is attached on the 3D printed exoskeleton. Hence, the compatibility issues for different body sizes should be addressed.
3. The basic working principle relies on the sliding mode of TENG, the authors need to discuss about the reliability of these sensors for long-term usage.
4. For the demonstration of force estimation, the kinetic analysis applied to the rotational sensing data can help to predict the straight punch forces without additional sensors. Can this approach be adopted for interpreting the data from other motions?
5. As the angle measurement is realized through the signal peaks generated when sliding across the gratings, what will happen if some of the gratings are damaged or cannot be detected?
6. Can the authors comment on the detailed advantages of this exoskeleton sensory system compare to those flexible wearable sensors? It would be better to have more details to get a clear understanding.
7. Please double check the figure numbers in supplementary materials (supplementary note 1 and 2).
8. Some other papers related to flexible wearable devices are also suggested to broad the horizon in the introduction part, such as:
[1] *Nature* 587, 219–224 (2020).
[2] *Science* 370, 961–965 (2020).
[3] *Research* 2020, 1–11 (2020).
[4] *npj Flex. Electron.* 4, 8 (2020).

Reviewer #2 (Remarks to the Author):

In this manuscript, authors report a kind of triboelectric bi-directional (TBD) sensing mechanism integrated with exoskeleton system for capturing and projecting the motions of human arm and finger into the robotic arm or virtual space. The working principle is clear, and the whole work is detailed and systematic. This work is very interesting with an innovation and shows a great potential of being an economic and advanced HMI for supporting the manipulation in both real and virtual

worlds, including robotic automation, healthcare, and training applications. I think the paper might be acceptable after a minor revision.

1. The shear force caused by rotation will twist the switch to deflect/rotate toward either clockwise or counter-clockwise direction and make the backside of switch touch E1 or E2, respectively. Whether the switch will affect the corresponding time of signal. Besides, the accuracy of sensors also can be mentioned in the manuscript.
2. Does the threshold voltage in Fig.3b change with the rotation speed? As shown in Fig2b (ii) and Fig 2c (ii), the output voltage will increase with the rotation speed. Hence, how to choose an appropriate threshold voltage in a non-uniform motion.
3. Compared with this work, the conventional solutions with the inertial and the resistive sensors have same drawbacks including the large power consumption. However, the amplifier/low pass filter/comparator and Arduino also need external energy supplies. And the energy consumption on the external circuit may be much greater than that of the resistance sensor. So, I personally think this advantage is unreasonable.
4. This TBD sensor can be used in ankle or neck motion? How about the environment influence on the sensor, such as humidity and temperature?

Reviewer #3 (Remarks to the Author):

The authors presented an interesting work, and demonstrated systematic applications of the proposed sensor. However, considering the overall precision, the feasibility and the sensor's durability, I could not suggest it to be published in the highly-reputed journal of Nature Communications.

1. Considering the working mechanism of triboelectric sensor, the triboelectric layer (PTFE films) in this work will be exposed to the air. The temperature and moisture, as well as any surrounding conducting objects, will have a large influence on the device output performance, which will apparently lower the sensor and system's SNR and stability.
2. The fabrication technique for the grating-patterned PTFE film shows a precision about 1 mm, and it is difficult to further decrease the electrode width and ensure the uniformity, including the size of the electrodes and gaps.
3. Additionally, owing to the fabrication technique, the device's resolution is about several degree. As shown in Fig. 2 b-c, the resolution is about $90 \text{ degree}/12$ (7.5 degree) or $140 \text{ degree}/12$ (11 degree). This performance need be essentially improved, to be accepted by the highly-reputed journal of Nature Communications.
4. Since the electrodes gap is about 3 mm, the response time of the sensor during moving forth and back would be long.
5. The tribo sensor's durability need be examined.

Dear Reviewers:

Your comments are all valuable and very helpful for revising and improving our manuscript. We have studied all of the comments carefully and made the revisions which we hope can meet with your approval. Briefly speaking, we have done extra experiments to collect new data, and added the additional discussion into the main manuscript in response to reviewers' comments. We have revised Fig. 2, supplementary Fig. 1, supplementary Fig. 6, and added supplementary Fig. 4-5. We also have added new references, include Ref. 26, 27, 28, 30, 33, 60 and 61. All of the revisions in the main manuscript and the supplementary information are marked in red, and the point-by-point response to the reviewers' comments are provided as following:

REVIEWER COMMENTS

Reviewer #1 (Remarks to the Author):

The authors propose a customized exoskeleton system with triboelectric bi-directional (TBD) sensors to realize the detection of both linear and rotational motions from human regarding joints rotation/twisting and finger bending. The switch used in TBD sensor offers great convenience to simplify the design of the grating patterns and the reading of the generated signals for performing the quantitative monitoring of bi-directional motions. In contrast to the recent research trend in the flexible wearable sensors (e.g., Nature 587, 219–224 (2020)), this exoskeleton based sensory system possesses its own superiority in the accurate detection of various motions. As a result, with the proper data analysis and the demonstrations with high completeness, the authors proved the dexterous motion projections in both VR and robotic manipulations, and also shown the possibilities of obtaining other kinematic parameters. This proposed exoskeleton sensing system illustrates a universal solution with various functionalities, which can be beneficial in diversified HMI applications. I think this work is very in time and do very comprehensive investigations on low-cost interface for human-robotic parallel control. Hence, I recommend the minor revision with the following comments.

Thanks a lot for reviewer's comment. The following is our point-to-point reply along with reviewer's comments.

1. As the rotational sensors, the authors demonstrated the sensing resolution of about 10 degree, and can be optimized via the design of grating patterns. Can the authors comment on the possible limit of resolution at current stage? And how about the actual sizes of these sensors?

We thank a lot for reviewer's valuable comment and suggestion. First of all, the actual size of each sensor is determined by the available space of the corresponding human joint, as well as the design of the grating patterns and switch. For a specific resolution, the size of the sensor

can be reduced if the dimensions of the grating patterns (width and spacing) can be reduced. In another word, for a fixed size, the reduced dimensions of the grating patterns can improve the resolution of the sensor. On the other hand, the available space of the corresponding human joint will define the general limitation of the largest size. In our proposed exoskeleton system, the diameters of the rotation sensors are varied from 6.5 cm (elbow) to 8.5 cm (wrist), according to the respective joint. All of these rotation sensors can be further miniaturized as shown in Supplementary Fig. 1, except RTBD-W, which is designed to allow the arm to insert.

In terms of sensing resolution, the smallest dimension of the grating patterns prepared in this manuscript is 1mm spacing and 1mm width. For a RTBD sensor with the diameter of 8.5 cm, the available gratings can provide the sensing resolution of about 4 degree. As mentioned above, the resolution of this dimension can be enhanced by increasing the size of the sensor. Furthermore, by utilizing the advanced fabrication processes, such as MEMS process, screen printing, micro-machining, printed circuit board, inkjet printing, etc., we can further reduce the dimensions of the gratings into μm level. Hence, the resolution can be further improved for a RTBD sensor with relative smaller size. This approach has been reported several times and proved to be a feasible and effective way to optimize the angular resolution of the triboelectric sensors under sliding mode. Jing et al. (Adv. Mater. Technol. 2018, 1800328) has reported an aerosol-jet printed fine-featured triboelectric angular sensor with an angular resolution about 2.5 degree¹. Wang et al. (Adv. Mater. 2020, 2001466) has utilized the printed circuit board technique to create the grating patterns, and achieved a nanoradian-resolution (1 degree)². However, in terms of bi-directional sensing, these reported devices require additional channel of the grating patterns to realize the detection of rotational direction by identifying the leading phase of among two channels. In our design, the single channel of grating pattern for bi-directional sensor can further reduce the space of sensing unit, and improve the resolution by eliminating the extra spacing of gratings for phase difference. We have added the relevant discussion into the main manuscript, and marked in red.

2. The whole sensory system is attached on the 3D printed exoskeleton. Hence, the compatibility issues for different body sizes should be addressed.

We thank a lot for reviewer's valuable comment and suggestion. Indeed, the compatibility issue of the size for wearable devices, especially for exoskeleton system, is very important for practical applications. The proposed exoskeleton is our preliminary design as a supportive platform for integrating our sensors. The optimization of this exoskeleton for better compatibility and more ergonomic is definitely needed. Specifically, for the current proposed device, we aimed at providing the 3D printed low cost solution of exoskeleton sensory system which can be self-fabricate and self-assembled according to the user's data.

On the other hand, for the massive production of the standard exoskeleton with certain compatibility, we do have two methods to solve this issue, so that the whole sensory system can fit different body sizes. As most of the sensors are located at the joints, we can redesign the middle part of upper arm and forearm exoskeleton to make them to have adjustable

length. For instance, the upper arm exoskeleton can be cut into two parts, and re-connected by a movable rail with a buckle (Fig. R1(a)), so that the length can be tuned and locked as needed. Secondly, we can replace the current design of inter-connected exoskeleton arm by the distributed joint armors, which are similar to the conventional rigid shoulder or elbow guard (Fig. R1(b)). Hence, the sensors can be separately attached onto the joints according to the specific requirement.

Fig. R1. Optimized designs for improving compatibility of exoskeleton sensory system. (a) Adjustable arm support by adding the sliding rail. (b) Elbow guard design for supporting the sensor.

We again appreciate the valuable suggestion. The corresponding discussion has been added into the description of the exoskeleton design part.

3. The basic working principle relies on the sliding mode of TENG, the authors need to discuss about the reliability of these sensors for long-term usage.

We thank a lot for reviewer's valuable comment and suggestion. The durability is a very important issue, especially for the sliding mode TENG sensor. To verify the influence of long-term usage on the signal quality, we have done the additional reliability test. As shown in Fig. R2, the test data of 3 hours is recorded (rotation speed: 100RPM), and there is no significant decreasing of the signal intensity, and the surface of PTFE layer does not have severe damage. This result can prove the robustness of the proposed sensor for long-term usage. Additionally, the peak counting method of angular sensing offers great advantage to the sensing signals which may experience the fluctuation of the amplitude. The whole rotation sensor can still operate even if it suffers the decay of the signal intensity after a long

period. On the other hand, the grating pattern of PTFE layer is designed as a replaceable modular part, which is featured with low cost and easy to replace. Hence, even if there is a need for maintenance, it will be very quick and convenient. We have added this reliability test into the supplementary information as supplementary Fig. 4.

Fig. R2. (i) Reliability test of the output signals for long-term operation of 3 hours. (ii) Output signals recorded at the beginning of the test. (iii) Output signals recorded at the end of the test.

4. For the demonstration of force estimation, the kinetic analysis applied to the rotational sensing data can help to predict the straight punch forces without additional sensors. Can this approach be adopted for interpreting the data from other motions?

We thank a lot for reviewer's valuable comment and suggestion. This is a good point for exploring the potentials of the proposed sensors. As the different parts of human body are connected via the corresponding joints, the various actions will actually create their own kinetic chain to perform the whole motions sequentially. Therefore, it is true that we can

fully analyze and utilize the obtained sensing information to estimate the different physical parameters during the motions, and rebuild the kinetic chain systematically. In our punching demonstration, we have done the primary investigation of this kinetic analysis to indicate the feasibility. The rotational sensing data can provide the linear displacement and the force information without introducing other type of sensors. This approach will significantly reduce the data complexity and the required computing power. Additionally, in terms of data interpretation for other motions, we can also obtain the linear acceleration via the second derivative of the displacement function. By further developing the lower limb exoskeleton, after collecting the dimensions of the lower limbs (length, mass, etc.), the walking pace, the walking speed, the step length, and even the stepping force are able to be estimated via the proper analysis of the sensing data from the sensors at ankles, knees, and hips, etc. We have added the relevant discussion into the main manuscript, and marked in red.

5. As the angle measurement is realized through the signal peaks generated when sliding across the gratings, what will happen if some of the gratings are damaged or cannot be detected?

We thank a lot for reviewer's valuable comment and suggestion. This is another important issue related to the reliability of the proposed sensor. It is true that the grating pattern will experience the damage issue during sliding motion. For the reported researches mentioned above (Adv. Mater. Technol. 2018, 1800328, Adv. Mater. 2020, 2001466), they were using the phase difference of the sliding generated peaks from the multiple channels to determine the rotation direction (i.e., “1212121212” for clockwise rotation of 5 degrees, and “21212121” for counterclockwise rotation of 4 degrees, “1” and “2” stand for signal peak from channel 1 and 2). For this type of designs, the damage of any channel may affect the identification of the leading phase, and cause the loss of direction sensing.

On the other hand, for our proposed sensor, we applied a single channel with the basic grating pattern to define the rotation angle, and a switch to determine the direction. Each grating can operate individually. Therefore, the damage of some gratings will not affect the general function of rotation sensing, but only cause the certain loss of synchronization, which can be partially solved by reprogramming the sensing algorithm based on the available grating number, or completely solved by replacing the modular grating pattern only.

Moreover, if the problem is some gratings cannot be detected, the spring design on the switch can offer a certain tunability for making the firm contact between two triboelectrification layers, and hence, the intensity of the signal can be improved for better detection.

6. Can the authors comment on the detailed advantages of this exoskeleton sensory system compare to those flexible wearable sensors? It would be better to have more details to get a clear understanding.

We thank a lot for reviewer's valuable comment and suggestion. The flexible wearable sensor is a very important and popular topic which consists of multiple research fields, such as

piezoresistive, capacitive, piezoelectric, triboelectric, EMG, etc. The advancements of the flexible sensors greatly improve the comfortability, the sensing functionalities, and the capability of integration with our common wearables (cloth, shoes, glove, etc.). Both physical and chemical sensors were frequently reported. Especially for flexible physical sensors, various parameters can be detected, including pressure, strain, bending/rotation, vibration, etc. However, most of those sensors are not able to provide the quantitative measurement of those physical parameters. Because the flexible sensors are usually attached on the irregular surface, and encountered with unstable motion. The soft body is also vulnerable to the unwanted impacts. Meanwhile, it is also difficult to ensure the uniformity of the as-fabricated flexible sensor, especially for large area deployment.

Therefore, our proposed exoskeleton system as a rigid sensing system, possesses several advantages compare to the flexible wearable sensors. Firstly, the sensors can offer reliable and quantitative monitoring of the human motions. Secondly, the single design of the switch and the grating pattern can realize the multi-functional and bi-directional sensing, including rotation, twisting, and linear motion. Thirdly, owing to the quantified sensing data, it allows the kinetic analysis for obtaining the additional physical parameters without adding other types of sensors.

7. Please double check the figure numbers in supplementary materials (supplementary note 1 and 2).

We thank a lot for reviewer's careful check. We have changed the figure numbers accordingly.

8. Some other papers related to flexible wearable devices are also suggested to broad the horizon in the introduction part, such as:

[1] Nature 587, 219–224 (2020).

[2] Science 370, 961–965 (2020).

[3] Research 2020, 1–11 (2020).

[4] npj Flex. Electron. 4, 8 (2020).

We thank a lot for reviewer's valuable comment and suggestion. The suggested papers were added into the introduction part of the manuscript to have a broader view to the current research advancements.

Reviewer #2 (Remarks to the Author):

In this manuscript, authors report a kind of triboelectric bi-directional (TBD) sensing mechanism integrated with exoskeleton system for capturing and projecting the motions of human arm and

finger into the robotic arm or virtual space. The working principle is clear, and the whole work is detailed and systematic. This work is very interesting with an innovation and shows a great potential of being an economic and advanced HMI for supporting the manipulation in both real and virtual worlds, including robotic automation, healthcare, and training applications. I think the paper might be acceptable after a minor revision.

Thanks a lot for reviewer's comment. The following is our point-to-point reply along with reviewer's comments.

1. The shear force caused by rotation will twist the switch to deflect/rotate toward either clockwise or counter-clockwise direction and make the backside of switch touch E1 or E2, respectively. Whether the switch will affect the corresponding time of signal. Besides, the accuracy of sensors also can be mentioned in the manuscript.

We thank a lot for reviewer's valuable comment and suggestion. The reviewer has mentioned a good point for bi-directional sensing. We have also considered this issue in our experiments. As shown in Fig. 2E, there are gaps between the edge of the switch and two electrodes, thus, the time of deflecting will affect the response time. Apparently, for a specific rotation speed, the larger gap will cause longer deflecting time. To further verify and evaluate this issue, we have designed two switches: one is normal edge, another is shaper edge, as shown in supplementary Fig. 4. The normal edge creates 3mm gaps, and the shaper edge gives 1.5mm gaps. As the latency of the response time is more severe at low speed, two rotation speed were tested: 10 RPM (low speed), and 100 RPM (normal speed). As shown in the collected data, the red peak indicates the left edge separate away from E1, and the black peak indicates the right edge contact with E2. Hence, the overall response time for a complete switching is the time difference between two peaks. As illustrated by the inset enlarged data graph of two peaks, there is only a small time difference of about a few ms, which indicates a relative good response time for switching the sensing direction. Interestingly, the 3mm gap also gives the similar result. The possible reason is the overall sizes of these two switches are all very small, and hence, the mm level gaps will not reveal a significant difference of the response time, even at very low rotation speed. There will be a noticeable advantage of having smaller gap only if the whole sensor has a very large size, which will not be applicable for our wearable system. We have added the response time and the accuracy/resolution of the proposed sensor into the main manuscript, and marked in red.

2. Does the threshold voltage in Fig.3b change with the rotation speed? As shown in Fig2b (ii) and Fig 2c (ii), the output voltage will increase with the rotation speed. Hence, how to choose an appropriate threshold voltage in a non-uniform motion.

We thank a lot for reviewer's valuable comment. As shown in Fig2b (ii) and Fig 2c (ii), the output voltage will increase as the rotation speed increases. Meanwhile, it is also worth to

mention that, the output intensity at low rotation speed (10 RPM) is already strong enough for back-end processing. Moreover, the basic sensing mechanism is to count the voltage peaks, rather than the amplitude of the peak. Therefore, we have set the threshold voltage based on the output voltage at lowest rotation speed, and this threshold voltage is then fixed for all of the demonstrations, without considering the variation of the rotation speed during the operation. For instance, if the peak value of output voltage is 0.5V, the threshold can be set as 0.3V, so that all of the voltage peaks above 0.3V will be counted as effective peak (a peak identification code is required to avoid the repeated counting caused by the data points within the incline and decline region above 0.3V of a single peak). Hence, we only need one relative low threshold voltage for peak identification of the non-uniform motions. On the other hand, although we want to keep the threshold voltage as low as possible, it is also necessary to ensure the threshold voltage is much higher than that of the background noises, to avoid the false counting. We have added this discussion into the main manuscript, and marked in red.

3. Compared with this work, the conventional solutions with the inertial and the resistive sensors have same drawbacks including the large power consumption. However, the amplifier/low pass filter/comparator and Arduino also need external energy supplies. And the energy consumption on the external circuit may be much greater than that of the resistance sensor. So, I personally think this advantage is unreasonable.

We thank a lot for reviewer's valuable comment and suggestion. This is a very good and popular question which remains quite debatable to the related research community. For those conventional inertial and resistive sensors, the great advancements of the researches during the past few decades assure the very low power consumption and high performance for the individual conventional sensor. On the other hand, the piezoelectric, triboelectric, thermoelectric, and other types of energy harvesting techniques are frequently reported with the sensing functions in recent years. Although there are several attempts to establish the complete self-sustained sensing system which are powered by those energy harvesters, most of the reported nanogenerator-based sensors are still requiring external power to support the signal processing circuit and the data transmission unit. Hence, in our opinion, we usually define our TENG sensor as zero-powered sensor, rather than self-powered sensor, in order to clarify that only the sensor itself is free from external power. In terms of the signal processing circuit, we assume that there is similar design and power consumption for both the conventional sensor and the zero-powered sensor. Although the individual power consumption of the conventional sensor itself may be very low, the massive distribution of sensors for a sensory network (such as the exoskeleton sensory system for capturing the full body motions) will eventually lead to higher total power consumption compare to the zero-powered sensor-based sensing network.

For the amplifier/low pass filter/comparator and Arduino used in our demonstration, these are the basic electronics which are applied in the prototype stage. Many of these components can be replaced by the commercial version with lower power consumption. The

microprocessor, Arduino, also has other alternatives, such as ESP32 integrated IoT module which requires much lower power, and has sleep mode to save the power. Moreover, some of the components can be removed, such as comparator, which can be easily realized via a simple code to identify the signal peaks.

Generally speaking, we totally understand the reviewer's concern about the advantage of power saving. The self-sustainability of the whole system is the ultimate goal and the essential requirement for the wearable and implantable system. As an interdisciplinary research field, it requires the continuous efforts from different expertise for the improvements of each component. Therefore, we have developed this exoskeleton sensory system with the zero-powered TENG sensor distributed at different joints of human body as a primary investigation. We have revised the relevant discussion in the introduction of the main manuscript, and marked in red.

4. This TBD sensor can be used in ankle or neck motion? How about the environment influence on the sensor, such as humidity and temperature?

We thank a lot for reviewer's valuable comment and suggestion. The reviewer understands the potential of TBD sensor very well. The key elements of the sensor are the switch design and the simple grating pattern. By customizing the sensor into the required structures (i.e., ring shape for arm joints, strip shape for finger bending), the TBD sensors can be applied for tracking the most of the human joint motions. In fact, we already have done the preliminary experiments for the lower limb, including ankle, knee and hip, etc., as shown in Fig. R3. As a result, the status of human walking can be detected and analyzed with the similar approach used in the sensing of the upper limb. These are ongoing works which will not be reported in this manuscript. For neck motion, it is also possible to be tracked. A backbone exoskeleton should be added as a main supporter, and a head frame is also needed. To connect these two parts, the connector with two degree of freedom is required, such as hooke joint, so that the two degree of freedom motions of the neck can be completely realized. The TBD sensors are then able to be integrated into the connector for tracking the neck motion.

Fig. R3. (i) Design of exoskeleton sensory system for lower limb. (ii) Demonstration of human machine interface.

The environment influences are indeed the critical parts need to be evaluated. To verify the reliability of the proposed sensor, we have done the extra experiments under the varied humidity and temperature, as shown in Fig. R4. Humidifier and hot air gun were applied to alter the humidity and temperature. All of these test setups, including temperature/humidity meter, step motor, and rotation sensor, were placed into a closed plastic container. For each test condition, the output measurement was started after the stabilization of 5 mins.

According to the experimental results, the humidity and temperature variation have effects on the intensity of the output signals. For our design, we adopted the peak counting method to realize the quantitative angular measurement. Hence, the variation of the signal intensity will not affect the performance of the proposed sensor, as long as the peak value still reach the threshold, which is also tunable. In general, even if the TENG sensor is experiencing a huge increasing of the humidity, the output voltages will not completely be vanished. The proper setting of the threshold voltage can ensure the enough tolerance to the humidity and the temperature variations. We have added the tests under varied humilities and temperatures into the supplementary information as supplementary Fig. 5.

Fig. R4. Comparison of the output signals under varied environmental conditions: (i) 20°C and 95% relative humidity, (ii) 20°C and 50 % relative humidity, (iii) 48°C and 20% relative humidity. Rotation speed: 100 RPM.

Reviewer #3 (Remarks to the Author):

The authors presented an interesting work, and demonstrated systematic applications of the proposed sensor. However, considering the overall precision, the feasibility and the sensor's durability, I could not suggest it to be published in the highly-reputed journal of Nature Communications.

Thanks a lot for reviewer's comment. The following is our point-to-point reply along with reviewer's comments.

1. Considering the working mechanism of triboelectric sensor, the triboelectric layer (PTFE films) in this work will be exposed to the air. The temperature and moisture, as well as any surrounding conducting objects, will have a large influence on the device output performance, which will apparently lower the sensor and system's SNR and stability.

We thank a lot for reviewer's valuable comment. The output fluctuation of triboelectric devices under different environment is indeed a common problem to the whole research community. The moisture in the air will affect the charge preservation on the triboelectrification layer, and hence, lower the peak amplitude of the rotation sensors in Fig. 2. The temperature and the conducting objects will also affect the output and bring the additional noises as well. To minimize these issues, the encapsulation techniques and the

functional materials were reported frequently, in order to improve the stability of TENG devices. For instance, Xiong et al. proposed hydrophobic cellulose oleoyl ester nanoparticles serves as a synergetic electron-trapping coating, rendering a textile nanogenerator with long-term reliability and high triboelectricity (Nat. Commun. 2018, 9:4280) 3. Jiang et al. reported an TENG tactile sensor array with flexible, transparent, and waterproof features for harsh operation environments (ACS Nano 2016, 10, 7696–7704) 4.

In our proposed exoskeleton sensory system, it is true that the primary design of the whole device is inevitably expose to the air. However, to minimize the environment influences, the basic sensing mechanism is designed to count the voltage peaks generated during the sliding of each grating, rather than measure the exact value of the peak. As shown in Fig. 2 and 3b, by sliding across a single grating, the sensor will generate a peak which will be converted into a unified square pulse (Fig. 3b) that represents the rotation of a certain degree. Hence, as long as each peak can reach the threshold voltage, the amplitude will not have any effect on the peak identification. For instance, by setting a threshold voltage of 0.3V, both 20V peak (at 300RPM) and 1V peak (at 50 RPM) will be identified as the effective peaks with the same information of rotation angle.

To further evaluate the temperature and humidity effects, we have done the extra experiments at the controlled environment conditions, as shown in Fig. R4. Humidifier and hot air gun were applied to change the conditions. All of these test setups, including temperature/humidity meter, step motor, and rotation sensor, were placed into a closed plastic container. For each test condition, the output measurement was started after the stabilization of 5 mins. As a result, even if the sensor is suddenly experiencing a high humidity (from 20% RH to 95% RH), and the output voltages drop significantly, the peak is still can be detected for generating the square pulse, since it still reaches the tunable threshold voltage. After the comparator circuit, there is no difference between the signals of 20% RH and the signals of 95% RH, as the control terminal is only receiving the label of peaks.

Fig. R4. Comparison of the output signals under varied environmental conditions: (i) 20°C and 95% relative humidity, (ii) 20°C and 50 % relative humidity, (iii) 48°C and 20% relative humidity. Rotation speed: 100 RPM.

In fact, our experiments and demonstration were all conducted at high relative humidity (70%~80% RH), all of the sensors were able to generate the signals with enough intensity. Hence, as the sensor is operated based on the peak counting method, the proper setting of the threshold voltage can ensure the enough tolerance to the humidity and the temperature variations.

Moreover, to minimize the noises caused by the surrounding conductive objects, we applied the additional ground channel (linked to the ground port of the microprocessor) which could be connected to human body (integrated with the glove in this design) to reduce the background noises generated by surrounding electronics and other conductive objects.

We have added the tests under varied humidities and temperatures into the supplementary information as supplementary Fig. 5.

2. The fabrication technique for the grating-patterned PTFE film shows a precision about 1 mm, and it is difficult to further decrease the electrode width and ensure the uniformity, including the size of the electrodes and gaps.

We thank a lot for reviewer's valuable comment. For the rotation sensor, the precision of 1mm (both the spacing and the width) for the ring with a diameter of 8.5 cm can provide the angular resolution of 4 degrees. In the early stage, the primary application scenarios of our proposed exoskeleton sensory system are to capture the human motions for realizing the human-machine interaction in virtual space and common humanoid robotic control. By considering the reasonable requirement of angular sensing resolution based on this low cost and facile designed sensor, we successfully achieved the general purposes of HMI. Alternatively, many applications of parallel control usually have feedback loop, such as vision inspection via our eyes. Thus, we can also adjust the ratio of motion projection to meet the requirements of precise control under the vision inspection, which is quite common in the setting of joystick and computer mouse sensitivity. For example, the sensor with the current resolution of 4 degrees can realize the control of 0.4 degree motion in virtual space or robot by programming a motion projection ratio of 10 : 1 into the code of the control terminal.

On the other hand, in terms of the fabrication techniques for better precision and uniformity, there are various well-established approaches to prepare the fine-featured gratings, such as such as MEMS process, screen printing, micro-machining, printed circuit board, inkjet printing, etc. The designs of the grating pattern for the linear or rotation sensing with sliding mode TENG sensor have been reported several times as shown in Fig. R5. The as-fabricated grating patterns can reach a feature size of micrometer level, while maintaining the enough signal intensity for peak detection. Jing et al. (Adv. Mater. Technol. 2018, 1800328) has reported an aerosol-jet printed fine-featured triboelectric angular sensor with an angular

resolution about 2.5 degree¹. Wang et al. (Adv. Mater. 2020, 2001466) has utilized the printed circuit board technique to create the grating patterns, and achieved a nanoradian-resolution (1 degree)². Therefore, we can leverage these techniques to realize the sensor with higher precision and uniformity, as well as reliability. However, this will also increase the cost of fabrication, especially for those customizable exoskeletons.

Fig. R5. (a) Aerosol-jet printed fine-featured triboelectric angular sensor¹. (b) Printed circuit board technique for achieved a nanoradian-resolution TENG sensor².

Similarly, the gaps of the switches are also able to be tuned accordingly. As shown in supplementary Fig. 4, the switch with the normal edge creates 3mm gaps, and the shaper edge gives 1.5mm gaps. After comparison of the response time, both of two design show similar result (a few milliseconds) at low rotation speed (10 RPM), since the overall sizes of two switches are small. In addition, the gaps can be further reduced by re-designing the structure of the edges.

In general, we strongly agree with the reviewer's viewpoint, the high precision and uniformity are crucial to some advanced applications. This is also the major task in our planned researches. The miniaturization and high precision fabrication process are well-established in the recent decades, so that we can leverage those available techniques to realize the fine-featured sensors. For the main novelties of our proposed exoskeleton sensory system, we are trying to emphasize on the introduction of the design of switch and the simple grating pattern as a universal solution for quantitative and multi-dimensional motion sensing, as well as the potential of estimating the multiple physical parameters via the kinetic analysis of the original rotation information. Eventually, this cost-effective and multi-functional sensor can be achieved without the complicate and multiple designs of sensing unit, in order to minimize the data and system complexity. Again, we appreciate that the reviewer addresses the concerns about the precision and the uniformity, and hope our reply can clarify these issues. We have added this discussion into the main manuscript, and marked in red.

3. Additionally, owing to the fabrication technique, the device's resolution is about several degree. As shown in Fig. 2 b-c, the resolution is about 90 degree/12 (7.5 degree) or 140 degree/12 (11 degree). This performance need be essentially improved, to be accepted by the highly-reputed journal of Nature Communications.

We thank a lot for reviewer's valuable comment and suggestion. This issue is also related to the previous comment about the precision of the grating pattern. In terms of sensing resolution, the 7.5 degrees resolution (90 degree/12) is actually calculated from the design of 1mm grating width and 3mm spacing. In Fig. 2 b-c (Fig. R6), the purpose of these characterization experiments with various parameters is to evaluate the influences of the specific variable on the sensing signals. Hence, we have tested the grating width of 1 mm, 3 mm, 5 mm, and 7 mm, as well as the grating spacing of 1 mm, 2 mm, and 3 mm. All of these designs can offer good sensing performance below a relative high rotation speed of 200 RPM. As a result, the smallest configuration of the grating patterns prepared in this manuscript can be 1 mm width and 1 mm spacing. In another word, all of the tested specifications in constant width test and constant spacing test can be recombined for designing the grating patterns, and we apologize for the vague description in the manuscript. For a RTBD sensor with the diameter of 8.5 cm (shoulder and wrist), the available gratings can provide the sensing resolution of about 4 degrees.

Fig. R6. b (i) Configuration of the varied grating widths (1 mm, 3mm, 5mm, 7mm) with a constant spacing of 3mm for a rotation TBD (RTBD) sensor, (ii) measurement of triboelectric output signals from the rotation speed of 10 RPM to 300 RPM, and (iii) the enlarged waveforms of 10 RPM and 300 RPM. c (i) Configuration of the varied spacing (1 mm, 2mm, 3mm) with a constant width of 3mm for a RTBD sensor. (ii) measurement of triboelectric output signals from the rotation speed of 10 RPM to 300 RPM, and (iii) the enlarged waveforms of 10 RPM and 200 RPM.

The resolution of this dimension can be enhanced by increasing the size of the sensor. On the other hand, by utilizing the advanced fabrication processes, such as MEMS process, screen printing, micro-machining, printed circuit board, inkjet printing, etc., we can further reduce the dimensions of the gratings into micrometer level, as mentioned before. Hence, the resolution can be further improved for a RTBD sensor with relative smaller size. By doing so, the switch design needs to be modified for ensuring the capability of detecting the fine-featured grating pattern and the signal stability. This is also the planned research for the next step of achieving the high precision parallel control in specialized HMI applications. At current stage, as the human body consists of multiple joints with various degree of freedoms, the joint-function of all the TBD sensors can still accomplish the majority of the daily tasks. Although the current device's resolution is about several degree. Besides, as mentioned before, we can also adjust the ratio of motion projection to meet the requirements of precise control.

We appreciate the reviewer's valuable criticism. This comment is highly in line with our research direction, and we will continue to bring more improvements in the exoskeleton sensory system. We have revised the corresponding part of characterization result in the main manuscript, and marked in red.

4. Since the electrodes gap is about 3 mm, the response time of the sensor during moving forth and back would be long.

We thank a lot for reviewer's valuable comment. As a clarification, to evaluate the response time for switching the sensing direction, we have designed two switches which gave the gaps of 1.5 mm and 3 mm. The switch used in the demonstration has 1.5 mm gap. It is true that the larger gap will cause longer deflecting time (response time) at a specific rotation speed. To further verify and evaluate this issue as shown in supplementary Fig. 4 (Fig. R7), the switch with the normal edge creates 3mm gaps, and the shaper edge gives 1.5mm gaps. Two rotation speed were tested: 10 RPM (low speed), and 100 RPM (normal speed). A complete switching process starts from one edge of the switch separate away from one electrode, till another edge contact with another electrode. To observe and capture these moments, we modified the electrodes by adding the PTFE triboelectrification layers, in order to create two small TENGs with contact-separation mode. As shown in the collected data, the red output peak indicates the left edge of the switch is separated away from E1, and the black peak indicates the right edge is contacted with E2. Hence, the response time for a complete switching is the time difference between two peaks. As illustrated by the inset enlarged data graph of two peaks, the time difference is about a few milliseconds, which indicates a relatively good response

time for switching the sensing direction. Interestingly, the 3mm gap also gives the similar result. The possible reason is the overall sizes of these two switches are all very small, and hence, the mm level gaps will not reveal a significant difference of the response time, even at very low rotation speed. In general, by considering the normal motion speed of human body, the response time of the sensor should be enough to ensure the synchronization during the parallel control. We have revised the corresponding part of the response time optimization of switch in the main manuscript and the supplementary information, and marked them in red.

Fig. R7. Evaluation of response latency during the switching of the direction for different designs of switches. a Switch with 1.5 mm gap at neutral state, and the corresponding signals from E1 and E2 electrodes during the switching, the red negative peak represents the separation of E1, and the black positive peak indicates the contact of E2, the enlarged graph of the dashed box, as well as the time and the peak voltages of the separation (1) and contact (2) signals are provided. b Switch with 1.5 mm gap at neutral state, and the corresponding signals from E1 and E2 electrodes during the switching, the enlarged graph of the dashed box as well as the time and the peak voltages of the separation (3) and contact (4) signals are provided.

5. The tribo sensor's durability need be examined.

We thank a lot for reviewer's valuable comment and suggestion. In terms of the durability of the sensor for long-term usage, we have done the additional reliability test. As shown in Fig. R2, the test data of 3 hours is recorded by microprocessor (Arduino) at a rotation speed of 100 RPM, and there is no significant decreasing of the signal intensity, and the surface of PTFE layer does not have severe damage. This result can prove the robustness of the proposed sensor for long-term usage. Additionally, similar to the humidity and temperature tests, the peak counting method of angular sensing offers great advantage to the sensing signals which may experience the fluctuation of the amplitude. The whole rotation sensor can still operate even if it suffers the decay of the signal intensity after a long period. On the other hand, the grating pattern of PTFE layer is designed as a replaceable modular part, which is featured with low cost and easy replacement. Hence, even if there is a need for maintenance, it will be very quick and convenient. We have added this reliability test into the supplementary information as supplementary Fig. 4.

Fig. R2. (i) Reliability test of the output signals recorded by microprocessor for long-term operation of 3 hours. (ii) Output signals recorded at the beginning of the test. (ii) Output signals recorded at the end of the test.

References:

1. Jing, Q. *et al.* Aerosol-Jet Printed Fine-Featured Triboelectric Sensors for Motion Sensing. *Adv. Mater. Technol.* **4**, 1800328 (2019).
2. Wang, Z. *et al.* A Self-Powered Angle Sensor at Nanoradian-Resolution for Robotic Arms and Personalized Medicare. *Adv. Mater.* **32**, 1–11 (2020).
3. Xiong, J. *et al.* Skin-touch-actuated textile-based triboelectric nanogenerator with black phosphorus for durable biomechanical energy harvesting. *Nat. Commun.* **9**, 4280 (2018).
4. Jiang, X. Z., Sun, Y. J., Fan, Z. & Zhang, T. Y. Integrated Flexible, Waterproof, Transparent, and Self-Powered Tactile Sensing Panel. *ACS Nano* **10**, 7696–7704 (2016).

Reviewer #1 (Remarks to the Author):

The authors have addressed all my comments, thus, I recommend to publish this manuscript at Nature Communications.

Reviewer #2 (Remarks to the Author):

The manuscript has been revised according to the reviewer's comments carefully. It has been improved. I recommend it to be accepted by Nature Communications.

REVIEWERS' COMMENTS

Reviewer #1 (Remarks to the Author):

The authors have addressed all my comments, thus, I recommend to publish this manuscript at Nature Communications.

We thank a lot for reviewer's recommendation and the great efforts in the previous review.

Reviewer #2 (Remarks to the Author):

The manuscript has been revised according to the reviewer's comments carefully. It has been improved. I recommend it to be accepted by Nature Communications.

We thank a lot for reviewer's recommendation and the great efforts in the previous review.